# Genome-wide meta-analysis of 241,258 adults accounting for smoking behaviour identifies novel loci for obesity traits

Anne E. Justice et al.[#]

Few genome-wide association studies (GWAS) account for environmental exposures, like smoking, potentially impacting the overall trait variance when investigating the genetic contribution to obesity-related traits. Here, we use GWAS data from 51,080 current smokers and 190,178 nonsmokers (87% European descent) to identify loci influencing BMI and central adiposity, measured as waist circumference and waist-to-hip ratio both adjusted for BMI. We identify 23 novel genetic loci, and 9 loci with convincing evidence of gene-smoking interaction (GxSMK) on obesity-related traits. We show consistent direction of effect for all identified loci and significance for 18 novel and for 5 interaction loci in an independent study sample. These loci highlight novel biological functions, including response to oxidative stress, addictive behaviour, and regulatory functions emphasizing the importance of accounting for environment in genetic analyses. Our results suggest that tobacco smoking may alter the genetic susceptibility to overall adiposity and body fat distribution.

#A full list of authors and their affiliations appears at the end of the paper.

Recent genome-wide association studies (GWAS) have described loci implicated in obesity, body mass index (BMI) and central adiposity. Yet most studies have ignored environmental exposures with possibly large impacts on the trait variance[1,2]. Variants that exert genetic effects on obesity through interactions with environmental exposures often remain undiscovered due to heterogeneous main effects and stringent significance thresholds. Thus, studies may miss genetic variants that have effects in subgroups of the population, such as smokers[3].

It is often noted that currently smoking individuals display lower weight/BMI and higher waist circumference (WC) as compared to nonsmokers[4–6]. Smokers also have the smallest fluctuations in weight over ~20 years compared to those who have never smoked or have stopped smoking[7,8]. Also, heavy smokers (>20 cigarettes per day [CPD]) and those that have smoked for more than 20 years are at greater risk for obesity than non-smokers or light to moderate smokers (<20 CPD)[9,10]. Men and women gain weight rapidly after smoking cessation and many people intentionally smoke for weight management[11]. It remains unclear why smoking cessation leads to weight gain or why long-term smokers maintain weight throughout adulthood, although studies suggest that tobacco use suppresses appetite[12,13] or alternatively, smoking may result in an increased metabolic rate[12,13]. Identifying genes that influence adiposity and interact with smoking may help us clarify pathways through which smoking influences weight and central adiposity[13].

A comprehensive study that evaluates smoking in conjunction with genetic contributions is warranted. Using GWAS data from the Genetic Investigation of Anthropometric Traits (GIANT) Consortium, we identified 23 novel genetic loci, and 9 loci with convincing evidence of gene-smoking interaction (GxSMK) on obesity, assessed by BMI and central obesity independent of overall body size, assessed by WC adjusted for BMI (WCadjBMI) and waist-to-hip ratio adjusted for BMI (WHRadjBMI). By accounting for smoking status, we focus both on genetic variants observed through their main effects and GxSMK effects to increase our understanding of their action on adiposity-related traits. These loci highlight novel biological functions, including response to oxidative stress, addictive behaviour and regulatory functions emphasizing the importance of accounting for environment in genetic analyses. Our results suggest that smoking may alter the genetic susceptibility to overall adiposity and body fat distribution.

## Results

**GWAS discovery overview.** We meta-analysed study-specific association results from 57 Hapmap-imputed GWAS and 22 studies with Metabochip, including up to 241,258 (87% European descent) individuals (51,080 current smokers and 190,178 nonsmokers) while accounting for current smoking (SMK) (Methods section, Supplementary Fig. 1, Supplementary Tables 1–4). For primary analyses, we conducted meta-analyses across ancestries and sexes. For secondary analyses, we conducted meta-analyses in European-descent studies alone and sex-specific meta-analyses (Tables 1–4, Supplementary Data 1–6). We considered four analytical approaches to evaluate the effects of smoking on genetic associations with adiposity traits (Fig. 1, Methods section). Approach 1 (SNPadjSMK) examined genetic associations after adjusting for SMK. Approach 2 (SNPjoint) considered the joint impact of main effects adjusted for SMK + interaction effects[14]. Approach 3 focused on interaction effects (SNPint); Approach 4 followed up loci from Approach 1 for interaction effects (SNPscreen). Results from Approaches 1–3 were considered genome-wide significant (GWS) with a $P$-value $< 5 \times 10^{-8}$ while Approach 4 used Bonferroni adjustment after screening. Lead variants >500 kb from

previous associations with BMI, WCadjBMI, and WHRadjBMI were considered novel. All association results are reported with effect estimates oriented on the trait increasing allele in the current smoking stratum.

Across the three adiposity traits, we identified 23 novel associated genetic loci (6 for BMI, 11 for WCadjBMI, 6 for WHRadjBMI) and nine having significant GxSMK interaction effects (2 for BMI, 2 for WCadjBMI, 5 for WHRadjBMI; Fig. 1, Tables 1–4, Supplementary Data 1–6). We provide a comprehensive comparison with previously-identified loci[1,2] by trait in supplementary material (Supplementary Data 7, Supplementary Note 1).

**Accounting for smoking status.** For primary meta-analyses of BMI (combined ancestries and sexes), 58 loci reached GWS in Approach 1 (SNPadjSMK; Supplementary Data 1, Supplementary Figs 2 and 3), including two novel loci near *SOX11*, and *SRRM1P2* (Table 1). Three more BMI loci were identified using Approach 2 (SNPjoint), including a novel locus near *CCDC93* (Supplementary Figs 4 and 5). For WCadjBMI, 62 loci reached GWS for Approach 1 (SNPadjSMK) and two more for Approach 2 (SNPjoint), including eight novel loci near *KIF1B*, *HDLBP*, *DOCK3*, *ADAMTS3*, *CDK6*, *GSDMC*, *TMEM38B* and *ARFGEF2* (Table 1, Supplementary Data 2, Supplementary Figs 2–5). Lead variants near *PSMB10* from Approaches 1 and 2 (rs14178 and rs113090, respectively) are >500 kb from a previously-identified WCadjBMI-associated variant (rs16957304); however, after conditioning on the known variant, our signal is attenuated ($P_{Conditional} = 3.02 \times 10^{-2}$ and $P_{Conditional} = 5.22 \times 10^{-3}$), indicating that this finding is not novel. For WHRadjBMI, 32 loci were identified in Approach 1 (SNPadjSMK), including one novel locus near *HLA-C*, with no additional loci in Approach 2 (SNPjoint; Table 1, Supplementary Data 3, Supplementary Figs 2–5).

We used GCTA[15] to identify loci from our primary meta-analyses that harbour multiple independent SNPs (Methods section, Supplementary Tables 5–7). Conditional analyses revealed no secondary signals within 500 kb of our novel lead SNPs. Additionally, we performed conditional association analyses to determine whether our novel variants were independent of previous GWAS loci within 500 kb that are associated with related traits of interest. All BMI-associated SNPs were independent of previously identified GWS associations with anthropometric and obesity-related traits. Seven novel loci for WCadjBMI were near previous associations with related anthropometric traits. Of these, association signals for rs6743226 near *HDLBP*, rs10269774 near *CDK6*, and rs6012558 near *ARFGEF2* were attenuated ($P_{Conditional} > 1E^{-5}$ and $\beta$ decreased by half) after conditioning on at least one nearby height and hip circumference adjusted for BMI (HIPadjBMI) SNP, but association signals remained independent of other related SNP-trait associations. For WHRadjBMI, our GWAS signal was attenuated by conditioning on two known height variants (rs6457374 and rs2247056), but remained significant in other conditional analyses. Given high correlations among waist, hip and height, these results are not surprising.

Several additional loci were identified for Approaches 1 and 2 in secondary meta-analysis (Table 2, Supplementary Data 1–6, Supplementary Fig. 6). For BMI, 2 novel loci were identified by Approach 1, including 1 near *EPHA3* and 1 near *INADL*. For WCadjBMI, 2 novel loci were identified near *RAI14* and *PRNP*. For WHRadjBMI, five novel loci were identified in secondary meta-analyses near *BBX*, *TRBI1*, *EHMT2*, *SMIM2* and *EYA4*. A comprehensive summary of nearby genes for all novel loci and their potential biological relevance is available in Supplementary Note 2.

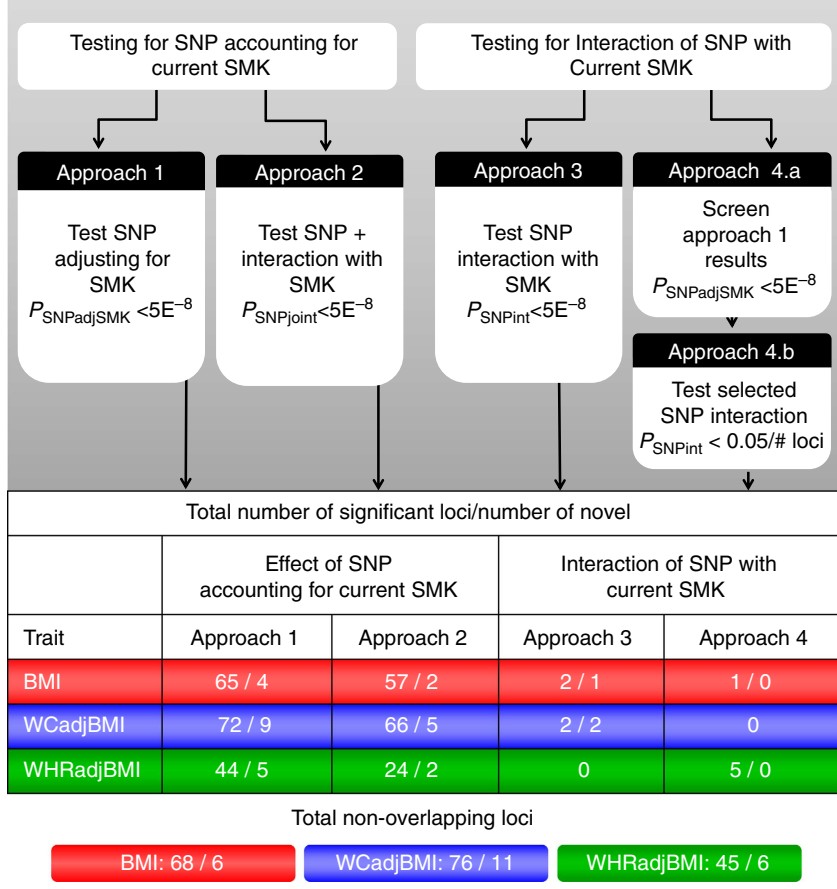

**Figure 1 | Summary of study design and results.** Approach 1 uses both SNP and SMK in the association model. Approaches 2 and 3 use the SMK-stratified meta-analyses. Approach 4 screens loci based on Approach 1, then uses SMK-stratified results to identify loci with significant interaction effects (Methods section).

Figure 3 presents analytical power for Approaches 1 and 2 while Supplementary Table 8 and Supplementary Fig. 7 present simulation results to evaluate type 1 error (Methods section). A heat map cross-tabulates $P$-values for Approaches 1 and 2 along with Approach 3 examining interaction only (Supplementary Fig. 8). We demonstrate that the two approaches yield valid type 1 error rates and that Approach 1 can be more powerful to find associations given zero or negligible quantitative interactions, whereas Approach 2 is more efficient in finding associations when interaction exists.

**Modification of genetic predisposition by smoking.** Approach 3 directly evaluated GxSMK interaction (SNPint; Table 3, Supplementary Data 1–6, Fig. 2, Supplementary Figs 9 and 10). For primary meta-analysis of BMI, two loci reached GWS including a previously identified GxSMK interaction locus near *CHRNB4* (ref. 3), and a novel locus near *INPP4B*. Both loci exhibit GWS effects on BMI in smokers and no effects in nonsmokers. For *CHRNB4* (cholinergic nicotine receptor B4), the variant minor allele (G) exhibits a decreasing effect on BMI in current smokers ($\beta$smk = − 0.047) but no effect in nonsmokers ($\beta$nonsmk = 0.002). Previous studies identified nearby SNPs in high LD associated with smoking (nonsynonymous, rs16969968 in *CHRNA5*)[3] and arterial calcification (rs3825807, a missense variant in *ADAMTS7*)[16]. Conditioning on these variants attenuated our interaction effect but did not eliminate it (Supplementary Table 7), suggesting a complex relationship between smoking, obesity, heart disease, and genetic variants in this region. Importantly, the *CHRNA5-CHRNA3-CHRNB4* gene

cluster has been associated with lower BMI in current smokers[3], but with higher BMI in never smokers[3], evidence supporting the lack of association in nonsmokers as well as a lack of previous GWAS findings on 15q25 (Supplementary Data 8)[1]. The *CHRNA5-CHRNA3-CHRNB4* genes encode the nicotinic acetylcholine receptor (nAChR) subunits α3, α5 and β4, which are expressed in the central nervous system[17]. Nicotine has differing effects on the body and brain, causing changes in metabolism and feeding behaviours[18]. These findings suggest smoking exposure may modify genetic effects on 15q24-25 to influence smoking-related diseases, such as obesity, through distinct pathways.

In primary meta-analyses of WCadjBMI, one novel GWS locus (near *GRIN2A*) with opposite effect directions by smoking status was identified for Approach 3 (SNPint; Table 3, Supplementary Data 2, Fig. 2, Supplementary Figs 9 and 10). The T allele of rs4141488 increases WCadjBMI in current smokers and decreases it in nonsmokers ($\beta$smk = 0.037, $\beta$nonsmk = − 0.015). In secondary meta-analysis of European women-only, we identified an interaction between rs6076699, near *PRNP*, and SMK on WCadjBMI (Table 4, Supplementary Data 5, Supplementary Fig. 6), a locus also identified in Approach 2 (SNPjoint) for European women. The major allele, A, has a positive effect on current smokers as compared to a weaker and negative effect on WC in nonsmokers ($\beta$smk = 0.169, $\beta$nonsmk = − 0.070), suggesting why this variant remained undetected in previous GWAS of WCadjBMI (Supplementary Data 8).

Approach 4 (SNPscreen; Fig. 1, Methods section) evaluated GxSMK interactions after screening SNPadjSMK results (from

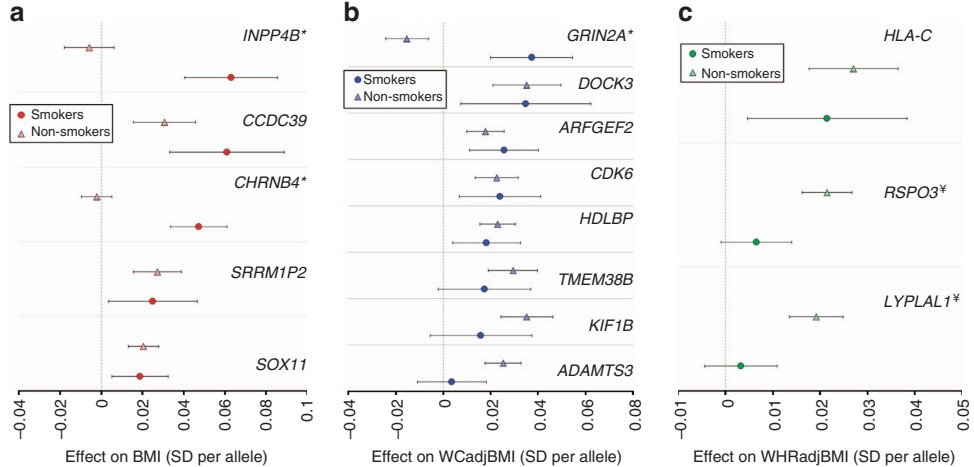

**Figure 2 | Forest plot for novel and GxSMK loci stratified by smoking status.** Estimated effects ($\beta \pm$ 95% CI) for smokers (N up to 51,080) and nonsmokers (N up to 190,178 ) per risk allele for (**a**) BMI, (**b**) WCadjBMI and (**c**) WHRadjBMI for novel loci from Approaches 1 and 2 (SNPadjSMK and SNPjoint, respectively) and all loci from Approaches 3 and 4 (SNPint and SNPscreen) identified in the primary meta-analyses. Loci are ordered by greater magnitude of effect in smokers compared to nonsmokers and labelled with the nearest gene. For the locus near *TMEM38B*, rs9409082 was used for effect estimates in this plot. (¥loci identified for Approach 4, *loci identified for Approach 3).

Approach 1) using Bonferroni-correction (Methods section, Tables 3–4, Supplementary Data 1–6). We identified two SNPs, near *LYPLAL1* and *RSPO3*, with significant interaction; both have previously published main effects on anthropometric traits. These loci exhibit effects on WHRadjBMI in nonsmokers, but not in smokers (Fig. 2). In secondary meta-analyses, we identified three known loci with significant GxSMK interaction effects on WHRadjBMI near *MAP3K1*, *HOXC4-HOXC6* and *JUND* (Table 4, Supplementary Data 3 and 6). We identified rs1809420, near *CHRNA5-CHRNA3-CHRNB4*, for BMI in the men-only, combined-ancestries meta-analysis (Supplementary Data 1).

Power calculations demonstrate that Approach 4 has increased power to identify SNPs that show (i) an effect in one stratum (smokers or nonsmokers) and a less pronounced but concordant effect in the other stratum, or (ii) an effect in the larger nonsmoker stratum and no effect in smokers (Fig. 3). In contrast, Approach 3 has increased power for SNPs that show (i) an effect in the smaller smoker stratum and no effect in nonsmokers, or (ii) an opposite effect between smokers and nonsmokers (Fig. 3). Our findings for both approaches agree with these power predictions, supporting using both analytical approaches to identify GxSMK interactions.

**Enrichment of genetic effects by smoking status**. When examining the smoking specific effects for BMI and WCadjBMI loci in our meta-analyses, no significant enrichment of genetic effects by smoking status were noted. (Fig. 2, Supplementary Figs 11 and 12). However, our results for WHRadjBMI were enriched for loci with a stronger effect in nonsmokers as compared to smokers, with 35 of 45 loci displaying numerically larger effects in nonsmokers ($P_{binomial} = 1.2 \times 10^{-4}$).

We calculated the variance explained by subsets of SNPs selected on 15 significance thresholds for Approach 1 from $P_{SNPadjSMK} = 1 \times 10^{-8}$ to $P_{SNPadjSMK} = 0.1$ (Supplementary Table 9, Fig. 4). Differences in variance explained between smokers and nonsmokers were significant ($P_{RsqDiff} < 0.003 = 0.05/15$, Bonferroni-corrected for 15 thresholds) for BMI at each threshold, with more variance explained in smokers. For WCadjBMI, the difference was significant for SNP sets beginning with $P_{SNPadjSMK} \geq 3.16 \times 10^{-4}$, and for WHRadjBMI at $P_{SNPadjSMK} \geq 1 \times 10^{-6}$. In contrast to BMI, SNPs from Approach 1 explained a greater proportion of the variance in nonsmokers

for WHRadjBMI. Differences in variance explained were greatest for BMI (differences ranged from 1.8 to 21% for smokers) and lowest for WHRadjBMI (ranging from 0.3 to 8.8% for nonsmokers).

These results suggest that smoking may increase genetic susceptibility to overall adiposity, but attenuate genetic effects on body fat distribution. This contrast is concordant with phenotypic observations of higher overall adiposity and lower central adiposity in smokers[4,6,7]. Additionally, smoking increases oxidative stress and general inflammation in the body[19] and may exacerbate weight gain[20]. Many genes implicated in BMI are involved in appetite regulation and feeding behaviour[1]. For waist traits, our results adjusted for BMI likely highlight distinct pathways through which smoking alters genetic susceptibility to body fat distribution. Overall, our results indicate that more loci remain to be discovered as more variance in the trait can be explained as we drop the threshold for significance.

**Functional or biological role of novel loci**. We conducted thorough searches of the literature and publicly available bioinformatics databases to understand the functional role of all genes within 500 kb of our lead SNPs. We systematically explored the potential role of our novel loci in affecting gene expression both with and without accounting for the influence of smoking behaviour (Methods section, Supplementary Note 3, Supplementary Tables 10–12).

We found the majority of novel loci are near strong candidate genes with biological functions similar to previously identified adiposity-related loci, including regulation of body fat/weight, angiogenesis/adipogenesis, glucose and lipid homeostasis, general growth and development. (Supplementary Notes 1 and 3).

We identified rs17396340 for WCadjBMI (Approaches 1 and 2), an intronic variant in the *KIF1B* gene. This variant is associated with expression of *KIF1B* in whole blood with and without accounting for SMK (GTeX and Supplementary Tables 10 and 12) and is highly expressed in the brain[21]. Knockout and mutant forms of *KIF1B* in mice resulted in multiple brain abnormalities, including hippocampus morphology[22], a region involved in (food) memory and cognition[23]. Variant rs17396340 is associated with expression levels of *ARSA* in LCL tissue. Human adipocytes express functional *ARSA*, which turns dopamine sulfate into active dopamine. Dopamine regulates appetite through leptin

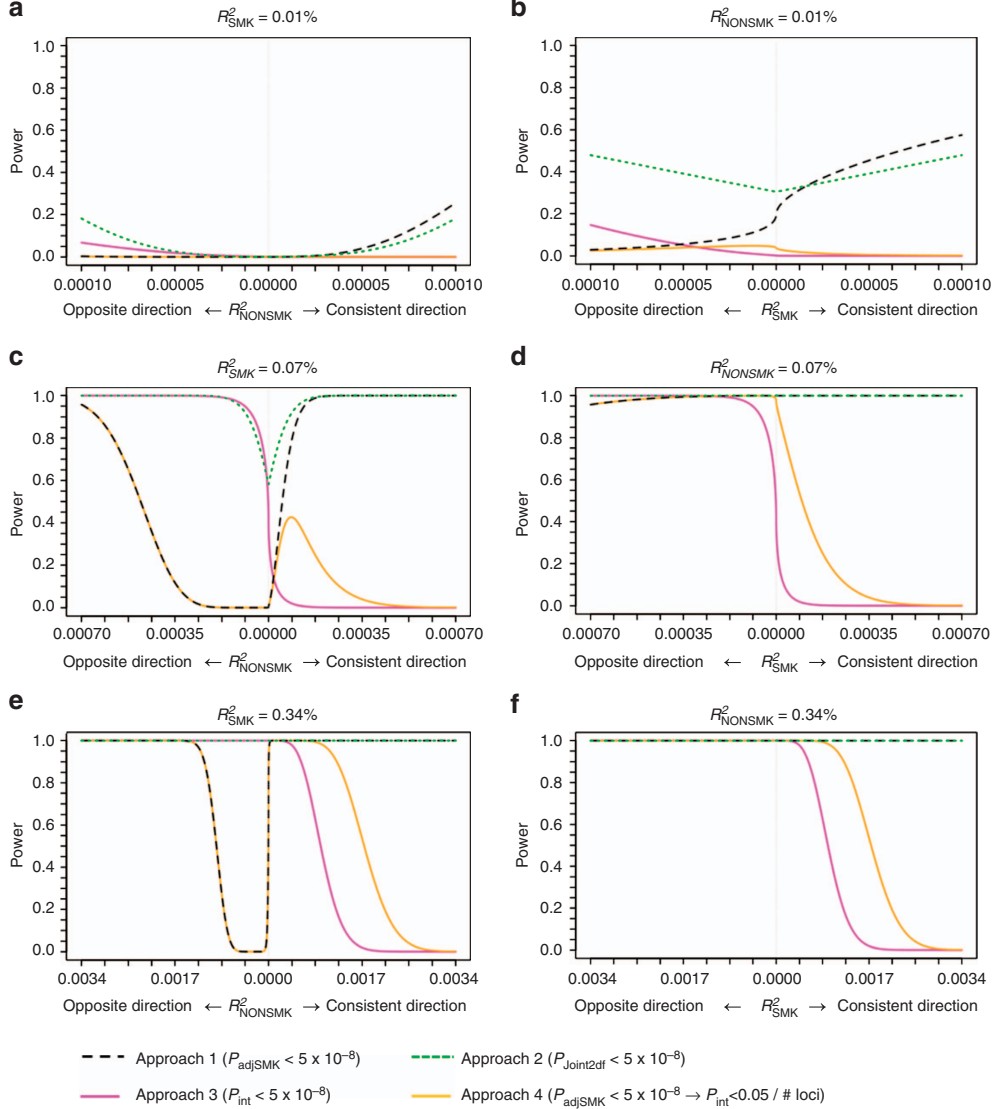

**Figure 3 | Power comparison across Approaches.** Shown is the power to identify adjusted (Approach 1, dashed black lines), joint (Approach 2, dotted green lines) and interaction (Approach 3 and 4, solid magenta and orange lines) effects for various combinations of SMK- and NonSMK-specific effects and assuming 50,000 smokers and 180,000 nonsmokers. For (**a,c,e**), the effect in smokers was fixed at a small ($R^2_{SMK} = 0.01\%$, similar to the realistic *NUDT3* effect on BMI), medium ($R^2_{SMK} = 0.07\%$, similar to the realistic *BDNF* effect on BMI) or large ($R^2_{SMK} = 0.34\%$, similar to the realistic *FTO* effect on BMI) genetic effect, respectively, and varied in nonsmokers. For (**b,d,f**), the effect in nonsmokers was fixed to the small, medium and large BMI effects, respectively, and varied in smokers.

and adiponectin levels, suggesting a role for *ARSA* in regulating appetite[24].

Expression of *CD47* (CD47 molecule), near rs670752 for WHRadjBMI (Approach 1, women-only), is significantly decreased in obese individuals and negatively correlated with BMI, WC and Hip circumference[25]. Conversely, in mouse models, CD47-deficient mice show decreased weight gain on high-fat diets, increased energy expenditure, improved glucose profile and decreased inflammation[26].

Several novel loci harbour genes involved in unique biological functions and pathways including addictive behaviours and response to oxidative stress. These potential candidate genes near our association signals are highly expressed in relevant tissues for regulation of adiposity and smoking behaviour (for example, brain, adipose tissue, liver, lung and muscle; Supplementary Note 2, Supplementary Table 10).

The *CHRNA5-CHRNA3-CHRNB4* cluster is involved in the eNOS signalling pathway (Ingenuity KnowledgeBase,

http://www.ingenuity.com) that is key for neutralizing reactive oxygen species introduced by tobacco smoke and obesity[27]. Disruption of this pathway has been associated with dysregulation of adiponectin in adipocytes of obese mice, implicating this pathway in downstream effects on weight regulation[27,28]. This finding is especially important due to the compounded stress adiposity places on the body as it increases chronic oxidative stress itself[28]. *INPP4B* has been implicated in the regulation of the PI3K/Akt signalling pathway[29] that is important for cellular growth and proliferation, but also eNOS signalling, carbohydrate metabolism, and angiogenesis[30].

*GRIN2A*, near rs4141488, controls long-term memory and learning through regulation and efficiency of synaptic transmission[31] and has been associated with heroin addiction[32]. Nicotine increases the expression of *GRIN2A* in the prefrontal cortex in murine models[33]. There are no established relationships between *GRIN2A* and obesity-related phenotypes in the literature, yet memantine and ketamine, pharmacological antagonists of

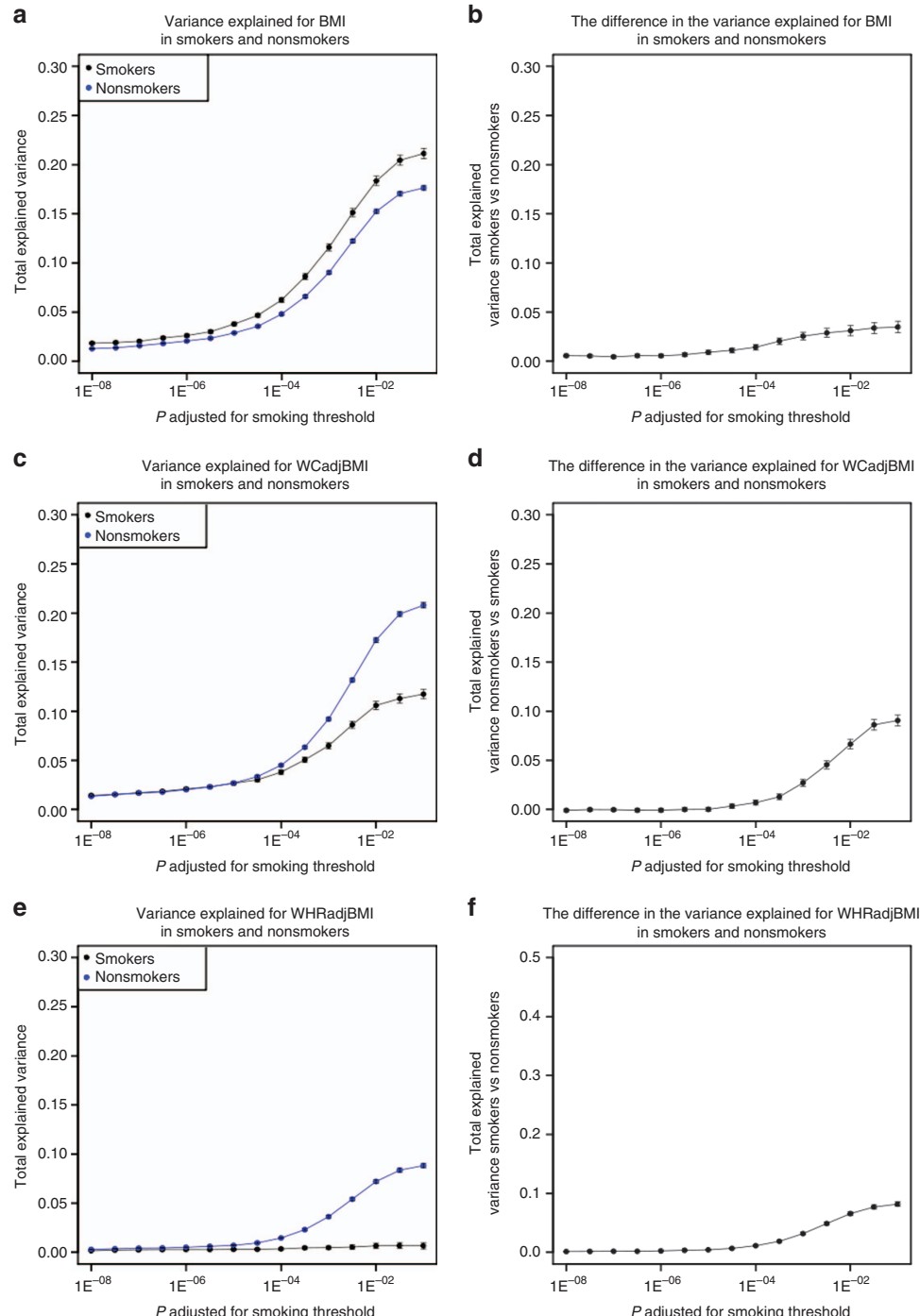

**Figure 4 | Stratum specific estimates of variance explained.** Total smoking status-specific explained variance (± s.e.) by SNPs meeting varying thresholds of overall association in Approach 1 (SNPadjSMK) and the difference between the proportion of variance explained between smokers and nonsmokers for these same sets of SNPs in BMI (**a,b**), WCadjBMI (**c,d**), and for WHRadjBMI (**e,f**).

GRIN2A activity[34,35], are implicated in treatment for obesity-associated disorders, including binge-eating disorders and morbid obesity (ClinicalTrials.gov identifiers: NCT00330655, NCT02334059, NCT01997515, NCT01724983). Memantine is under clinical investigation for treatment of nicotine dependence (ClinicalTrials.gov identifiers: NCT01535040, NCT00136786 and NCT00136747). While our lead SNP is not within a characterized gene, rs4141488 and variants in high LD ($r^2 > 0.7$) are within active enhancer regions for several tissues, including liver, fetal leg muscle, smooth stomach and intestinal muscle, cortex and several embryonic and pluripotent cell types (Supplementary Note 2),

and therefore may represent an important regulatory region for nearby genes like *GRIN2A*.

In secondary meta-analysis of European women-only, we identified a significant GxSMK interaction for rs6076699 on WCadjBMI (Table 4, Supplementary Data 4, Supplementary Fig. 6). This SNP is 100 kb upstream of *PRNP* (prion protein), a signalling transducer involved in multiple biological processes related to the nervous system, immune system, and other cellular functions (Supplementary Note 2)[36]. Alternate forms of the oligomers may form in response to oxidative stress caused by copper exposure[37]. Copper is present in cigarette smoke and

**Table 1 | Summary of association results for novel loci reaching genome-wide significance in Approach (App) 1 ($P_{SNPadjSMK}$ $<5E^{-8}$) or Approach 2 ($P_{SNPjoint}$ $<5E^{-8}$) for our primary meta-analysis in combined ancestries and combined sexes.**

| App | Marker | Chr:Pos (hg19) | Nearest Gene | N | EAF | Alleles E/O | Smokers | | Non-smokers | | Main and interaction effects | | | | GIANT + UKBB | | |
|---|---|---|---|---|---|---|---|---|---|---|---|---|---|---|---|---|---|
| | | | | | | | $\beta$ | P-value | $\beta$ | P-value | $\beta_{adj}$ | $P_{SNPadjSMK}$ | $P_{SNPint}$ | $P_{SNPjoint}$ | $P_{SNPadjSMK}$ | $P_{SNPint}$ | $P_{SNPjoint}$ |
| *BMI* | | | | | | | | | | | | | | | | | |
| 1,2 | rs10929925 | 2:6155557 | SOX11 | 225,067 | 0.55 | C/A | 0.019 | 7.80E⁻03 | 0.02 | 8.40E⁻08 | 0.020 | **1.1E⁻09** | 8.2E⁻01 | **1.6E⁻08** | **1.5E⁻13** | 4.5E⁻01 | **9.8E⁻13** |
| 1 | rs6794880 | 3:84451512 | SRRM1P2 | 186,968 | 0.85 | A/G | 0.025 | 2.30E⁻02 | 0.027 | 3.90E⁻06 | 0.028 | **4.3E⁻08** | 8.5E⁻01 | 1.8E⁻06 | **4.9E⁻09** | 4.5E⁻01 | 9.7E⁻08 |
| 2 | rs13069244 | 3:180441172 | CCDC39 | 233,776 | 0.08 | A/G | 0.061 | 1.80E⁻05 | 0.031 | 6.60E⁻05 | 0.035 | 1.2E⁻07 | 4.6E⁻02 | **3.5E⁻08** | **6.1E⁻10** | 1.1E⁻02 | **9.6E⁻11** |
| *WCadjBMI* | | | | | | | | | | | | | | | | | |
| 1,2 | rs17396340 | 1:10286176 | KIF1B | 206,485 | 0.14 | A/G | 0.016 | 1.40E⁻01 | 0.035 | **4.70E⁻10** | 0.028 | **3.0E⁻08** | 9.8E⁻01 | **9.1E⁻10** | **1.0E⁻11** | 2.9E⁻02 | **1.5E⁻13** |
| 1,2 | rs6743226 | 2:242236972 | HDLBP | 200,666 | 0.53 | C/T | 0.018 | 1.30E⁻02 | 0.023 | **2.60E⁻09** | 0.023 | **1.2E⁻10** | 5.5E⁻01 | **5.8E⁻10** | **6.7E⁻12** | 7.0E⁻01 | **2.8E⁻11** |
| 1 | rs4378999 | 3:51208646 | DOCK3 | 156,566 | 0.13 | T/A | 0.035 | 1.30E⁻02 | 0.035 | 1.30E⁻06 | 0.036 | **4.1E⁻08** | 9.7E⁻01 | 4.1E⁻07 | **7.6E⁻11** | 5.3E⁻01 | **3.2E⁻10** |
| 1,2 | rs7697556 | 4:73515313 | ADAMTS3 | 206,017 | 0.49 | T/C | 0.004 | 6.30E⁻01 | 0.025 | **7.30E⁻11** | 0.021 | **5.2E⁻09** | 6.7E⁻03 | **7.6E⁻10** | **5.4E⁻19** | 1.9E⁻01 | **2.7E⁻19** |
| 1 | rs10269774 | 7:92253972 | CDK6 | 157,552 | 0.34 | A/G | 0.024 | 6.60E⁻03 | 0.023 | 1.10E⁻06 | 0.023 | **2.9E⁻08** | 8.8E⁻01 | 1.6E⁻07 | **2.9E⁻10** | 7.7E⁻01 | **2.1E⁻09** |
| 1 | rs6470765 | 8:130736697 | GSDMC | 157,450 | 0.76 | A/C | 0.032 | 1.90E⁻03 | 0.023 | 1.70E⁻05 | 0.026 | **4.8E⁻08** | 4.3E⁻01 | 9.5E⁻07 | **2.5E⁻12** | 8.9E⁻01 | **9.0E⁻11** |
| 2 | rs9408815 | 9:108890521 | TMEM38B | 156,427 | 0.75 | C/G | 0.012 | 2.30E⁻01 | 0.03 | **4.20E⁻09** | 0.03 | **2.3E⁻08** | 8.5E⁻01 | **1.7E⁻08** | **1.2E⁻11** | 3.0E⁻01 | **2.8E⁻11** |
| 1 | rs9409082 | 9:108901049 | | 157,785 | 0.76 | C/T | 0.017 | 8.10E⁻02 | 0.029 | **2.60E⁻08** | 0.027 | **1.5E⁻08** | 2.7E⁻01 | **4.6E⁻08** | **9.5E⁻12** | 6.6E⁻01 | **6.5E⁻11** |
| 1 | rs6012558 | 20:47531286 | ARFGEF2 | 208,004 | 0.41 | A/G | 0.026 | 5.40E⁻04 | 0.018 | 6.50E⁻06 | 0.020 | **1.9E⁻08** | 3.3E⁻01 | 1.3E⁻07 | **1.5E⁻09** | 7.0E⁻02 | **3.0E⁻09** |
| *WHRadjBMI* | | | | | | | | | | | | | | | | | |
| 1,2 | rs1049281 | 6:31236567 | HLA-C | 149,285 | 0.66 | C/T | 0.022 | 1.30E⁻02 | 0.027 | **2.00E⁻08** | 0.025 | **2.2E⁻09** | 5.6E⁻01 | **5.3E⁻09** | **1.2E⁻18** | 8.3E⁻01 | **1.8E⁻10** |

Adj, adjusted for smoking; app, approach; int, interaction; chr, chromosome; EAF, effect allele frequency; E/O, effect/other; Pos, position (bp). Significant P-values that reach genome-wide significance ($P < 5 \times 10^{-8}$) threshold are in bold.

elevated in the serum of smokers, but is within safe ranges[38,39]. Another gene near rs6076699, *SLC23A2* (Solute Carrier Family 23 (Ascorbic Acid Transporter), Member 2), is essential for the uptake and transport of Vitamin C, an important nutrient for DNA and cellular repair in response to oxidative stress both directly and through supporting the repair of Vitamin E after exposure to oxidative agents[40,41]. SLC23A2 is present in the adrenal glands and murine models indicate that it plays an important role in regulating dopamine levels[42]. This region is associated with success in smoking cessation and is implicated in addictive behaviours in general[43,44]. Our tag SNP is located within an active enhancer region (marked by open chromatin marks, DNAse hypersentivity, and transcription factor binding motifs); this regulatory activity appears tissue specific (sex-specific tissues and lungs; HaploReg and UCSC Genome Browser).

Nicotinamide mononucleotide adenylyltransferease (*NMNAT1*), upstream of WCadjBMI variant rs17396340, is responsible for the synthesis of NAD from ATP and NMN[45]. NAD is necessary for cellular repair following oxidative stress. Upregulation of *NMNAT* protects against damage caused by reactive oxygen species in the brain, specifically the hippocampus[46]. Also for WCadjBMI, both *CDK6*, near SNP rs10269774, and *FAM49B*, near SNP rs6470765, are targets of the *BACH1* transcription factor, involved in cellular response to oxidative stress and management of the cell cycle[47].

**Influence of novel loci on related traits.** In a look-up in existing GWAS of smoking behaviours (Ever/Never, Current/Not-Current, Smoking Quantity (SQ))[48] (Supplementary Data 8), eight of our 26 SNPs were nominally associated with at least one smoking trait. After multiple test correction ($P_{Regression} < 0.05/26 = 0.0019$), only one SNP remains significant: rs12902602, identified for Approaches 2 (SNPjoint) and 3 (SNPint) for BMI, showed association with SQ ($P = 1.45 \times 10^{-9}$).

We conducted a search in the NHGRI-EBI GWAS Catalog[49,50] to determine if any of our newly identified loci are in high LD with variants associated with related cardiometabolic and behavioural traits or diseases. Of the seven novel BMI SNPs, only rs12902602 was in high LD ($r^2 > 0.7$) with SNPs previously associated with smoking-related traits (for example, nicotine dependence), lung cancer, and cardiovascular diseases (for example, coronary heart disease; Supplementary Table 13). Of the 12 novel WCadjBMI SNPs, 5 were in high LD with

previously reported GWAS variants for mean platelet volume, height, infant length, and melanoma. Of the six novel WHRadjBMI SNPs, three were near several previously associated variants, including cardiometabolic traits (for example, LDL cholesterol, triglycerides and measures of renal function).

Given high phenotypic correlation between WC and WHR with height, and established shared genetic associations that overlap our adiposity traits and height[1,2,51] we expect cross-trait associations between our novel loci and height. Therefore, we conducted a look-up of all of our novel SNPs to identify overlapping association signals (Supplementary Data 8). No novel BMI loci were significantly associated with height ($P_{Regression} < 0.002(0.05/24)$ SNPs). However, there are additional variants that may be associated with height, but not previously reported in GWAS examining height, including two for WHRadjBMI near *EYA4* and *TRIB1*, and two for WCadjBMI near *KIF1B* and *HDLBP* ($P_{Regression} < 0.002$).

Finally, as smoking has a negative (weight decreasing) effect on BMI, it is likely that smoking-associated genetic variants have an effect on BMI in current smokers. Therefore, we expected that smoking-associated SNPs exhibit some interaction with smoking on BMI. We looked up published smoking behaviour SNPs[49,50], 10 variants in 6 loci, in our own results. Two variants reached nominal significance ($P_{SNPint} < 0.05$) for GxSMK interaction on BMI (Supplementary Table 14), but only one reached Bonferroni-corrected significance ($P < 0.005$). No smoking-associated SNPs exhibited GxSMK interaction. Therefore, we did not see a strong enrichment for low interaction $P$ values among previously identified smoking loci.

**Validation of novel loci.** We pursued validation of our novel and interaction SNPs in an independent study sample of up to 119,644 European adults from the UK Biobank study (Tables 1–4, Supplementary Table 15, Supplementary Fig. 9). We found consistent directions of effects in smoking strata (for Approaches 2 and 3) and in SNPadjSMK results (Approach 1) for each locus examined (Supplementary Fig. 13). For BMI, three SNPs were not GWS ($P_{SNPadjSMK}$, $P_{SNPjoint}$, $P_{SNPInt} > 5E^{-8}$) following meta-analysis with our GIANT results: rs12629427 near *EPAH3* (Approach 1); rs1809420 within a known locus near *ADAMTS7* (Approach 4) remained significant for interaction, but not for SNPadjSMK; and rs336396 near *INPP4B* (Approach 3). For WCadjBMI, 3 SNPs were not GWS ($P_{SNPadjSMK}$, $P_{SNPjoint}$, $P_{SNPInt} > 5E^{-8}$) following meta-analysis with our results:

**Table 2 | Novel loci showing significant association in Approaches 1 (SNPadjSMK) and/or 2 (SNPjoint) identified in secondary meta-analyses and not significant in primary meta-analyses.**

| Approach: Strata | Marker | Chr:Pos (hg19) | Nearest Gene | N | EAF | Alleles E/O | Smokers β | Smokers P-value | Non-smokers β | Non-smokers P-value | $\beta_{adj}$ | $P_{SNPadj}$ | $P_{SNPint}$ | $P_{SNPjoint}$ | $P_{SNPadjSMK}$ | $P_{SNPint}$ | $P_{SNPjoint}$ |
|---|---|---|---|---|---|---|---|---|---|---|---|---|---|---|---|---|---|
| **BMI** | | | | | | | | | | | | | | | | | |
| 1:EC | rs2481665 | 1:62594677 | INADL | 209,453 | 0.56 | T/C | 0.015 | $4.60\times10^{-2}$ | 0.021 | $8.90\times10^{-8}$ | 0.019 | **$3.50\times10^{-8}$** | $4.00\times10^{-1}$ | $6.70\times10^{-8}$ | **$3.3\times10^{-11}$** | $7.8\times10^{-1}$ | **$2.0\times10^{-8}$** |
| 1:AW | rs12629427 | 3:89145340 | EPHA3 | 137,961 | 0.26 | C/T | 0.025 | $2.10\times10^{-2}$ | 0.028 | $3.60\times10^{-7}$ | 0.027 | **$4.80\times10^{-8}$** | $8.00\times10^{-1}$ | $2.00\times10^{-7}$ | **$7.7\times10^{-8}$** | $9.1\times10^{-1}$ | $3.0\times10^{-7}$ |
| 1:EW | rs2173039 | 3:89142175 | | 117,942 | 0.26 | C/G | 0.024 | $3.10\times10^{-2}$ | 0.032 | $8.90\times10^{-8}$ | 0.031 | **$7.30\times10^{-9}$** | $5.70\times10^{-1}$ | $6.50\times10^{-8}$ | **$2.4\times10^{-9}$** | $9.3\times10^{-1}$ | $2.2\times10^{-7}$ |
| **WCadjBMI** | | | | | | | | | | | | | | | | | |
| 1:EM | rs1545348 | 5:34718343 | RAI14 | 77,677 | 0.73 | T/G | 0.044 | $3.10\times10^{-4}$ | 0.03 | $1.90\times10^{-5}$ | 0.034 | **$1.80\times10^{-8}$** | $3.20\times10^{-1}$ | $1.70\times10^{-7}$ | $1.2\times10^{-7}$ | $1.2\times10^{-1}$ | $4.8\times10^{-7}$ |
| 2:EW | rs6076699 | 20:4566688 | PRNP | 76,930 | 0.97 | A/G | 0.169 | $1.40\times10^{-5}$ | −0.07 | $1.20\times10^{-4}$ | −0.034 | $3.50\times10^{-2}$ | **$1.40\times10^{-8}$** | **$4.80\times10^{-8}$** | $4.2\times10^{-2}$ | $2.3\times10^{-6}$ | $3.4\times10^{-6}$ |
| **WHRadjBMI** | | | | | | | | | | | | | | | | | |
| 1:AW | rs670752 | 3:107312980 | BBX | 107,568 | 0.32 | A/G | 0.012 | $5.50\times10^{-2}$ | 0.009 | $1.50\times10^{-2}$ | 0.027 | **$4.90\times10^{-8}$** | $6.80\times10^{-1}$ | $7.80\times10^{-3}$ | **$3.1\times10^{-10}$** | $3.8\times10^{-1}$ | $9.5\times10^{-5}$ |
| 1:EC | rs589428 | 6:31848220 | EHMT2 | 162,918 | 0.66 | G/T | 0.006 | $1.20\times10^{-1}$ | 0.011 | $4.10\times10^{-4}$ | 0.022 | **$2.80\times10^{-8}$** | $3.50\times10^{-1}$ | $7.00\times10^{-4}$ | **$1.1\times10^{-17}$** | $8.4\times10^{-2}$ | **$1.6\times10^{-10}$** |
| 2:EC | rs1856293 | 6:133480940 | EYA4 | 127,431 | 0.52 | A/C | 0.006 | $5.30\times10^{-1}$ | −0.028 | $9.10\times10^{-9}$ | −0.019 | $6.50\times10^{-6}$ | $5.40\times10^{-1}$ | **$4.70\times10^{-8}$** | $9.6\times10^{-8}$ | $1.3\times10^{-2}$ | **$1.5\times10^{-8}$** |
| 1:AW | rs2001945 | 8:126477978 | TRIB1 | 103,446 | 0.4 | G/C | 0.009 | $1.20\times10^{-1}$ | 0.013 | $1.00\times10^{-4}$ | 0.025 | **$4.70\times10^{-8}$** | $5.90\times10^{-1}$ | $1.30\times10^{-4}$ | **$1.1\times10^{-9}$** | $3.0\times10^{-1}$ | $1.4\times10^{-6}$ |
| 1:EC | rs17065323 | 13:44627788 | SMIM2* | 69,968 | 0.01 | T/C | 0.154 | $1.90\times10^{-1}$ | −0.23 | $1.20\times10^{-10}$ | −0.181 | **$9.20\times10^{-9}$** | $1.40\times10^{-3}$ | **$3.90\times10^{-10}$** | **$9.6\times10^{-9}$** | $3.6\times10^{-3}$ | **$1.3\times10^{-9}$** |

A, all ancestries; C, combined sexes; Chr, chromosome; E, European-only; EAF, effect allele frequency; E/O, effect/other; int, interaction; M, men only; Pos, position (bp); Padj, adjusted for smoking; W, women only.
All estimates are from the stratum specified in the Approach:Sample column.
*This locus was filtered from approaches 2–4 due to low sample size in the SMK strata, and only P values for Approach 1 are considered significant. Significant P-values that reach genome-wide significance ($P < 5 \times 10^{-8}$) threshold are in bold.

**Table 3 | Summary of association results for loci showing significance for interaction with smoking in Approach (App) 3 (SNPint) and/or Approach 4 (SNPscreen) in our primary meta-analyses of combined ancestries and combined sexes.**

| App | Marker | Chr:Pos (hg19) | Nearest Gene | N | EAF | Alleles E/O | Smokers β | Smokers P-value | Non-smokers β | Non-smokers P-value | $\beta_{adj}$ | $P_{SNPadj}$ | $P_{SNPint}$ | $P_{SNPjoint}$ | $P_{SNPadjSMK}$ | $P_{SNPint}$ | $P_{SNPjoint}$ |
|---|---|---|---|---|---|---|---|---|---|---|---|---|---|---|---|---|---|
| **BMI** | | | | | | | | | | | | | | | | | |
| 3 | rs336396 | 4:143062811 | INPP4B | 169,646 | 0.18 | T/C | 0.063 | **$4.8\times10^{-8}$** | −0.006 | $3.4\times10^{-1}$ | 0.007 | $2.3\times10^{-1}$ | **$2.1\times10^{-8}$** | $1.9\times10^{-7}$ | $7.4\times10^{-1}$ | $2.7\times10^{-6}$ | $1.3\times10^{-5}$ |
| 3 | rs12902602* | 15:78967401 | CHRNB4 | 240,135 | 0.62 | A/G | 0.047 | **$1.8\times10^{-11}$** | −0.002 | $5.5\times10^{-1}$ | 0.009 | $8.6\times10^{-3}$ | **$4.1\times10^{-11}$** | **$1.1\times10^{-10}$** | $1.1\times10^{-1}$ | **$6.0\times10^{-13}$** | **$1.6\times10^{-12}$** |
| **WCadjBMI** | | | | | | | | | | | | | | | | | |
| 3 | rs4141488 | 16:9629067 | GRIN2A | 153,892 | 0.5 | T/C | 0.037 | $2.2\times10^{-5}$ | −0.015 | $9.6\times10^{-4}$ | −0.003 | $4.4\times10^{-1}$ | **$2.7\times10^{-8}$** | $5.0\times10^{-7}$ | $9.5\times10^{-1}$ | $1.8\times10^{-6}$ | $1.1\times10^{-5}$ |
| **WHRadjBMI** | | | | | | | | | | | | | | | | | |
| 4 | rs765751* | 1:219669226 | LYPLAL1 | 189,028 | 0.64 | C/T | 0.003 | $3.9\times10^{-1}$ | 0.019 | **$3.1\times10^{-11}$** | 0.029 | **$3.1\times10^{-16}$** | $7.3\times10^{-4}$ | **$2.1\times10^{-10}$** | **$9.1\times10^{-31}$** | $1.4\times10^{-4}$ | **$7.8\times10^{-22}$** |
| 4 | rs7766106* | 6:127455138 | RSPO3 | 188,174 | 0.48 | T/C | 0.007 | $7.9\times10^{-2}$ | 0.022 | **$2.2\times10^{-15}$** | 0.037 | **$3.7\times10^{-27}$** | $9.7\times10^{-4}$ | **$3.8\times10^{-15}$** | **$4.4\times10^{-51}$** | $1.0\times10^{-5}$ | **$3.4\times10^{-34}$** |

Adj, adjusted for smoking; app, approach; int, interaction; chr, chromosome; EAF, effect allele frequency; E/O, effect/other; Pos, position (bp).
*Known locus.
Significant P-values after multiple test correction are italicized.
Significant P-values that reach genome-wide significance ($P < 5 \times 10^{-8}$) threshold are in bold.

rs1545348 near *RAI14* (Approach 1); rs4141488 near *GRIN2A* (Approach 3); and rs6012558 near *PRNP* (Approach 3). For WHRadjBMI, only 1 SNP from Approach 4 was not significant following meta-analysis with our results: rs12608504 near *JUND* remained GWS for SNPadjSMK, but was only nominally significant for interaction ($P_{SNPint} = 0.013$).

**Challenges in accounting for environmental exposures in GWAS.** A possible limitation of our study may be the definition and harmonization of smoking status. We chose to stratify on current smoking status without consideration of type of smoking (for example, cigarette, pipe) for two reasons. First, focusing on weight alone, former smokers tend to return to their expected weight quickly following smoking cessation[7,13,52]. Second, this definition allowed us to maximize sample size, as many participating studies only had current smoking status available. However, WC and WHR may not behave in the same manner as weight and BMI with former smokers retaining excess fat around their waist. Thus, results may differ with alternative harmonization of smoking exposure.

Another limitation may be potential bias in our effect estimates when adjusting for a correlated covariate (for example, collider bias)[53]. This phenomenon is of particular concern when the correlation between the outcome and the covariate is high and when significant genetic associations occur with both traits in opposite directions. Our analyses adjusted both WC and WHR for BMI. WHR has a correlation of 0.49 with BMI, while WC has

a correlation of 0.85 (ref. 53). Using previously published results for BMI, WCadjBMI and WHRadjBMI, we find three novel loci for WCadjBMI (near *DOCK3*, *ARFGEF2* and *TMEM38B*) and two for WHRadjBMI (near *EHMT2* and *HLA-C*; Supplementary Data 8) with nominally significant associations with BMI and opposite directions of effect. At these loci, the genetic effect estimates should be interpreted with caution. Additionally, we adjusted for SMK in Approach 1 (SNPadjSMK). However binary smoking status, as we used, has a low correlation to BMI, WC, and WHR, as estimated in the ARIC study's European descent participants (−0.13, 0.08 and 0.12, respectively) and in the Framingham Heart Study (−0.05, 0.08 and 0.16). Additionally, there are no loci identified in Approach 1 (SNPadjSMK) that are associated with any smoking behaviour trait and that exhibit an opposite direction of effect from that identified in our adiposity traits (Supplementary Data 8). We therefore preclude potential collider bias and postulate true gain in power through SMK-adjustment at these loci.

To assess how much additional information is provided by accounting for SMK and GxSMK in GWAS for obesity traits, we compared genetic risk scores (GRSs) based on various subsets of lead SNP genotypes in various regression models (Methods section). While any GRS was associated with its obesity trait ($P_{GRS} < 1.6 \times 10^{-7}$, Supplementary Table 16), adding SMK and GxSMK terms to the regression model along with novel variants to the GRSs substantially increased variance explained. For example, variance explained increased by 38% for BMI (from

**Table 4 | Summary of association results for loci showing significance for interaction with smoking in Approach 3 (SNPint) and/or Approach 4 (SNPscreen) in our secondary meta-analyses not identified in primary meta-analyses.**

| Approach: Strata | Marker | Chr:Pos (hg19) | Nearest Gene | N | EAF | Alleles E/O | Smokers | | Non-smokers | | Main and interaction effects | | | | GIANT + UKBB | | |
|---|---|---|---|---|---|---|---|---|---|---|---|---|---|---|---|---|---|
| | | | | | | | $\beta$ | $P$ | $\beta$ | $P$ | $\beta_{adj}$ | $P_{SNPadj}$ | $P_{SNPint}$ | $P_{SNPjoint}$ | $P_{SNPadjSMK}$ | $P_{SNPint}$ | $P_{SNPjoint}$ |
| *BMI* | | | | | | | | | | | | | | | | | |
| 4:AM | rs1809420* | 15:79056769 | *ADAMTS7* | 57,081 | 0.59 | T/C | 0.074 | **$9.8\times10^{-08}$** | 0.023 | $2.0\times10^{-03}$ | 0.036 | **$4.9\times10^{-08}$** | $9.4\times10^{-04}$ | **$5.6\times10^{-09}$** | $9.8\times10^{-05}$ | *$3.3\times10^{-05}$* | $1.9\times10^{-07}$ |
| *WCadjBMI* | | | | | | | | | | | | | | | | | |
| 3:EW | rs6076699 | 20:4566688 | *PRNP* | 76,930 | 0.97 | A/G | 0.169 | $1.4\times10^{-05}$ | −0.07 | $1.2\times10^{-04}$ | −0.034 | $3.5\times10^{-02}$ | **$1.4\times10^{-08}$** | **$4.8\times10^{-08}$** | $4.2\times10^{-02}$ | $2.3\times10^{-06}$ | $3.4\times10^{-06}$ |
| *WHRadjBMI* | | | | | | | | | | | | | | | | | |
| 4:EM | rs30000* | 5:55803533 | *MAP3K1* | 71,424 | 0.27 | G/A | 0.002 | $7.8\times10^{-01}$ | 0.031 | $3.7\times10^{-08}$ | 0.04 | **$1.7\times10^{-10}$** | $1.6\times10^{-04}$ | $2.7\times10^{-07}$ | **$2.7\times10^{-17}$** | *$3.2\times10^{-07}$* | **$3.8\times10^{-15}$** |
| 4:AM | rs459193* | 5:55806751 | | 80,852 | 0.27 | A/G | 0.004 | $5.0\times10^{-01}$ | 0.034 | **$4.1\times10^{-10}$** | 0.043 | **$2.3\times10^{-13}$** | $6.8\times10^{-05}$ | **$2.2\times10^{-09}$** | **$3.5\times10^{-20}$** | *$2.5\times10^{-07}$* | **$1.6\times10^{-17}$** |
| 4:AM | rs2071449* | 12:54428011 | *HOXC4-* | 70,868 | 0.37 | A/C | 0.003 | $6.0\times10^{-01}$ | 0.026 | $1.0\times10^{-06}$ | 0.034 | **$9.1\times10^{-09}$** | $1.1\times10^{-03}$ | $5.7\times10^{-06}$ | **$2.7\times10^{-12}$** | $8.0\times10^{-04}$ | **$2.8\times10^{-09}$** |
| 4:EM | rs754133* | 12:54418920 | *HOXC6* | 71,136 | 0.36 | A/G | 0.003 | $6.2\times10^{-01}$ | 0.026 | $8.2\times10^{-07}$ | 0.034 | **$3.0\times10^{-09}$** | $1.1\times10^{-03}$ | $4.0\times10^{-06}$ | **$2.1\times10^{-12}$** | $9.7\times10^{-04}$ | **$4.0\times10^{-09}$** |
| 4:AM | rs12608504* | 19:18389135 | *JUND* | 80,087 | 0.37 | A/G | 0.006 | $2.6\times10^{-01}$ | 0.025 | $5.0\times10^{-07}$ | 0.032 | **$4.7\times10^{-09}$** | $5.5\times10^{-03}$ | $1.8\times10^{-06}$ | **$2.9\times10^{-11}$** | $1.3\times10^{-02}$ | **$1.6\times10^{-08}$** |

A, all ancestries; Adj, adjusted for smoking; app, approach; C, combined sexes; Chr, chromosome; E, European-only; EAF, effect allele frequency; E/O, effect/other; int, interaction; M, men only; Pos, position (bp); W, women only.
All estimates are from the stratum specified in the Approach:Sample column The $R^2$ between the *ADAMTS7* (rs1809420) and *CHRNB4* variant (rs1290362) in Table 3 is 0.72 (HapMap 2, CEU).
Additionally, the *PRNP* variant (rs6076699) is the same as the variant that came up from Approach 2 (Table 2).
*Known locus.
Significant *P*-values after multiple test correction are italicized.
Significant *P*-values that reach genome-wide significance ($P < 5 \times 10^{-8}$) threshold are in bold.

1.53 to 2.11%, $P_{GRSDiff} = 4.3 \times 10^{-5}$), by 27% for WCadjBMI (from 2.59 to 3.29%, $P_{GRSDiff} = 3.9 \times 10^{-6}$) and by 168% for WHRadjBMI (from 0.82 to 2.20%, $P_{GRSDiff} = 3.2 \times 10^{-11}$). Therefore, despite potential limitations, much is gained by accounting for environmental exposures in GWAS studies.

## Discussion

To better understand the effects of smoking on genetic susceptibility to obesity, we conducted meta-analyses to uncover genetic variants that may be masked when the environmental influence of smoking is not considered, and to discover genetic loci that interact with smoking on adiposity-related traits. We identified 161 loci in total, including 23 novel loci (6 for BMI, 11 for WCadjBMI, and 6 for WHRadjBMI). While many of our newly identified loci support the hypothesis that smoking may influence weight fluctuations through appetite regulation, these novel loci also have highlighted new biological processes and pathways implicated in the pathogenesis of obesity.

Importantly, we identified nine loci with convincing evidence of GxSMK interaction on obesity-related traits. We were able to replicate the previous GxSMK interaction with BMI within the *CHRNA5-CHRNA3-CHRNB4* gene cluster. One novel BMI-associated locus near *INPP4B* and two novel WCadjBMI-associated loci near *GRIN2A* and *PRNP* displayed significant GxSMK interaction. We were also able to identify significant GxSMK interaction for one known BMI-associated locus near *ADAMTS7* and for five known WHRadjBMI-associated loci near *LYPLAL1*, *RSPO3*, *MAP3K1*, *HOXC4-HOXC6* and *JUND*. The majority of these loci harbour strong candidate genes for adiposity with a possible role for the modulation of effects through tobacco use.

We identified 18 new loci in Approach 1 ($P_{SNPadjSMK}$) by adjusting for current smoking status. Our analyses did not allow us to determine whether these discoveries are due to different subsets of subjects included in the analyses compared to previous studies[1,2] or due only to adjusting for current smoking. Adjustment for current smoking in our analyses, however, did reveal novel associations. Specifically after accounting for smoking in our analyses, all novel BMI loci exhibit *P*-values that are at least one order of magnitude lower than in previous GIANT investigations, despite smaller samples in the current analysis[2]. While sample sizes for both WCadjBMI and WHRadjBMI are comparable with previous GIANT investigations, our *P* values for variants identified in Approach 1 are at least two orders of magnitude lower than previous findings. Thus, adjustment for smoking may have indeed revealed new loci. Further, loci identified in Approach 2, including nine

novel loci, suggest that accounting for interaction improves our ability to detect these loci even in the presence of only modest evidence of GxSMK interaction.

There are several challenges in validating genetic associations that account for environmental exposure. In addition to exposure harmonization and potential bias due to adjustment for smoking exposure, differences in trait distribution, environmental exposure frequency, ancestry-specific LD patterns and allele frequency across studies may lead to difficulties in replication, especially for gene-by-environment studies[54]. Furthermore, the 'winner's curse' (inflated discovery effects estimates) requires larger sample sizes for adequate power in replication studies[55]. Despite these challenges, we were able to detect consistent direction of effect in an independent sample for all novel loci. Some results that did not remain GWS in the GIANT + UKBB meta-analysis had results that were just under the threshold for significance, suggesting that a larger sample may be needed to confirm these results, and thus the associations near *INPP4B*, *GRIN2A*, *RAI14*, *PRNP* and *JUND* should be interpreted with caution.

While we found that effects were not significantly enriched in smokers for BMI, there is a greater proportion of variance in BMI explained by variants that are significant for Approach 1 (SNPadjSMK), which may be expected given that there are a greater number of variants with higher effect estimates in smokers. For WCadjBMI, there was no enrichment for stronger effects in one stratum compared to the other for our significant loci; however, there was a greater proportion of explained variance in WCadjBMI for loci identified in Approach 1 (SNPadjSMK) in nonsmokers. For WHRadjBMI, there were significantly more loci that exhibit greater effects in nonsmokers, and this pattern was mirrored in the variance explained analysis. The large difference between effects in smokers and nonsmokers likely explains the sub-GWS levels of our loci in previous GIANT investigations[2]. For example, the T allele of rs7697556, 81kb from the *ADAMTS3* gene, was associated with increased WCadjBMI and exhibits a sixfold greater effect in nonsmokers compared to smokers, although the interaction effect was only nominal; in previous GWAS this variant was nearly GWS. These differences in effect estimates between smokers and nonsmokers may help explain inconsistent findings in previous analyses that show central adiposity increases with increased smoking, but is associated with decreased weight and BMI[5,9,10].

Our results support previous findings that implicate genes involved in transcription and gene expression, appetite regulation, macronutrient metabolism, and glucose homeostasis. Several of our novel loci have candidate genes within 500 kb of our tag

variants that are highly expressed and/or active in brain tissue (*BBX, KIF1B, SOX11* and *EPHA3*) and, like other obesity-associated genes, may be involved in previously-identified pathways linked to neuronal regulation of appetite (*KIF1B, GRIN2A* and *SLC23A2*), adipo/angiogenesis (*ANGPTL3* and *TNF*) and glucose, lipid and energy homeostasis (*CD47, STK25, STK19, RAGE, AIF1, LYPLAL1, HDLBP, ANGPTL3, DOCK7, KIF1B, PREX1* and *RPS12*).

Many our newly identified loci highlight novel biological functions and pathways where dysregulation may lead to increased susceptibility to obesity, including response to oxidative stress, addictive behaviour, and newly identified regulatory functions. There is a growing body of evidence that supports the notion that exposure to oxidative stress leads to increased adiposity, risk of obesity, and poor cardiometabolic outcomes[27,56]. Our results for BMI and WCadjBMI, specifically associations identified near *CHRNA5-CHRNA3-CHRNB4, PRNP, SLC23A2, BACH1* and *NMNAT1,* highlight new biological pathways and processes for future examination and may lead to a greater understanding of how oxidative stress leads to changes in obesity phenotypes and downstream cardiometabolic risk.

By considering current smoking, we were able to identify 6 novel loci for BMI, 11 for WCadjBMI and 6 for WHRadjBMI, and highlight novel biological processes and regulatory functions for genes implicated in increased obesity risk. Eighteen of these remained significant in our validation with the UK Biobank sample. We confirmed most established loci in our analyses after adjustment for smoking status in smaller samples than were needed in previous discovery analyses. A typical approach in large-scale GWAS meta-analyses is not to adjust for covariates such as current smoking; our findings highlight the importance of accounting for environmental exposures in genetic analyses.

## Methods

**Study design overview.** We applied four approaches to identify genetic loci that influence adiposity traits by accounting for current tobacco smoking status (Fig. 1). We defined smokers as those who responded that they were currently smoking; not current smokers were those that responded 'no' to currently smoking. We evaluated three traits: body mass index (BMI), waist circumference adjusted for BMI (WCadjBMI), and waist-to-hip ratio adjusted for BMI (WHRadjBMI). Our first two meta-analytical approaches were aimed at determining whether there are novel genetic variants that affect adiposity traits by adjusting for SMK (SNPadjSMK), or by jointly accounting for SMK and for interaction with SMK (SNPjoint); while Approaches 3 and 4 aimed to determine whether there are genetic variants that affect adiposity traits through interaction with SMK (SNPint and SNPscreen) (Fig. 1). Our primary meta-analyses focused on results from all ancestries, sexes combined. Secondary meta-analyses were performed using the European-descent populations only, as well as stratified by sex (men-only and women-only) in all ancestries and in European-descent study populations.

**Cohort descriptions and sample sizes.** The GIANT consortium was formed by an international group of researchers interested in understanding the genetic architecture of anthropometric traits (Supplemental Tables 1–4 for study sample sizes and descriptive statistics). In total, we included up to 79 studies comprising up to 241,258 individuals for BMI (51,080 smokers, 190,178 non-smokers), 208,176 for WCadjBMI (43,226 smokers, 164,950 non-smokers), and 189,180 for WHRadjBMI (40,543 smokers, 148,637 non-smokers) with HapMap II imputed genome-wide chip data (up to 2.8M SNPs in association analyses), and/or with genotyped MetaboChip data (~195 K SNPs in association analyses). In instances where studies submitted both Metabochip and GWAS data, these were for non-overlapping individuals. Each study's Institutional Review Board has approved this research and all study participants have provided written informed consent.

**Phenotype descriptions.** Our study highlights three traits of interest: BMI, WCadjBMI and WHRadjBMI. Height and weight, used to calculate BMI (kg m$^{-2}$), were measured in all studies; waist and hip circumferences were measured in the vast majority. For each sex, traits were adjusted using linear regression for age and age$^2$ (as well as for BMI for WCadjBMI and WHRadjBMI), and (when appropriate) for study site and principal components to account for ancestry. Family studies used linear mixed effects models to account for familial relationships and also conducted analyses for men and women combined including sex in the model. Phenotype residuals were obtained from the adjustment models

and were inverse normally transformed subsequently to facilitate comparability across studies and with previously published analyses. The trait transformation was conducted separately for smokers and nonsmokers for the SMK-stratified model and using all individuals for the SMK-adjusted model.

**Defining smokers.** The participating studies have varying levels of information on smoking, some with a simple binary variable and others with repeated, precise data. Since the effects of smoking cessation on adiposity appear to be immediate[7,8,52], a binary smoking trait (current smoker versus not current smoker) is used for the analyses as most studies can readily derive this variable. We did not use a variable of 'ever smoker vs. never' as it increases heterogeneity across studies, thus adding noise; also this definition would make harmonization across studies difficult.

**Genotype identification and imputation.** Studies with GWAS array data or Metabochip array data contributed to the results. Each study applied study-specific standard exclusions for sample call rate, gender checks, sample heterogeneity and ethnic group outliers (Supplementary Table 2). For each studies (except those that employed directly typed MetaboChip genotypes), genome-wide chip data was imputed to the HapMap II reference data set.

**Study level analyses.** To obtain study-specific summary statistics used in subsequent meta-analyses, the following linear models (or linear mixed effects models for studies with families/related individuals) were run separately for men and women and separately for cases and controls for case-control studies using phenotype residuals from the models described above. Studies with family data also conducted analyses with these models for men and women combined after accounting for dependency among family members as a function of their kinship correlations. We assumed an additive genetic model. The analyses were run using various GWAS software Supplementary Table 2.

$$\text{SMK-adjusted}: \quad \text{TRAIT} = \beta_0 + \beta_1 \text{SNP} + \beta_2 \text{SMK}$$
$$\text{SMK-stratified}: \quad \text{TRAIT} = \beta_0 + \beta_1 \text{SNP} \ (\text{run in current smokers and} \\ \text{nonsmokers separately})$$

**Quality control of study-specific summary statistics.** The aggregated summary statistics were quality-controlled according to a standardized protocol[57]. These included checks for issues with trait transformations, allele frequencies and strand. Low quality SNPs in each study were excluded for the following criteria: (i) SNPs with low minor allele count (MAC < = 5, MAC = MAF × N) and monomorphic SNPs, (ii) genotyped SNPs with low SNP call-rate (<95%) or low Hardy-Weinberg equilibrium test $P$ value (<10$^{-6}$), (iii) imputed SNPs with low imputation quality (MACH-Rsq or OEVAR <0.3, or information score <0.4 for SNPTEST/IMPUTE/IMPUTE2, or <0.8 for PLINK). To test for issues with relatedness or overlapping samples and to correct for potential population stratification, the study-specific standard errors and association $P$ values were genomic control (GC) corrected using lambda factors (Supplementary Fig. 1). GC correction for GWAS data used all SNPs, but GC correction for MetaboChip data were restricted to chip QT interval SNPs only as the chip was enriched for associations with obesity-related traits. Any study-level GWAS file with a lambda >1.5 was removed from further analyses. While we established this criterion, no study results were removed for this reason.

**Meta-analyses.** Meta-analyses used study-specific summary statistics for the phenotype associations for each of the above models. We used a fixed-effects inverse variance weighted method for the SNP main effect analyses. All meta-analyses were run in METAL[58]. As study results came in two separate batches (Stage 1 and Stage 2), meta-analyses from the two stages were further meta-analysed (Stage 1 + Stage 2). A second GC correction was applied to all SNPs when combining Stage 1 and Stage 2 meta-analyses in the final meta-analyses. First, Hapmap-imputed GWAS data were meta-analysed together, as were Metabochip studies. This step was followed by a combined GWAS + Metabochip meta-analysis. For primary analyses, we conducted meta-analyses across ancestries and sexes. For secondary meta-analyses, we conducted meta-analyses in European-descent studies alone, and sex-specific meta-analyses. There were two reasons for conducting secondary meta-analyses. First, both WCadjBMI and WHRadjBMI have been shown to display sex-specific genetic effects[2,59,60]. Second, by including populations from multiple ancestries in our primary meta-analyses, we may be introducing heterogeneity due to differences in effect sizes, allele frequencies, and patterns of linkage disequilibrium across ancestries, potentially decreasing power to detect genetic effects. See Supplementary Fig. 1 for a summary of the primary meta-analysis study design. The obtained SMK-stratified summary statistics were later used to calculate summary SNPjoint and SNPint statistics using EasyStrata[61]. Briefly, this software implements a two-sample, large sample test of equal regression parameters between smokers and nonsmokers[59] for SNPint and the two degree of freedom test of main and interaction effects for SNPjoint[14].

**Lead SNP selection.** Before selecting a lead SNP for each locus, SNPs with high heterogeneity $I^2 \geq 0.75$ or a minimum sample size below 50% of the maximum

$N$ for each strata (for example, $N > \max(N$ women smokers$)/2$) were excluded. Lead SNPs that met significance criteria were selected based on distance ($\pm 500$ kb), and we defined the SNP with the lowest $P$ value as the top SNP for a locus. SNPs that reached genome-wide significance (GWS), but had no other SNPs within 500 kb with a $P < 1E-5$ (lonely SNPs), were excluded from the SNP selection process. Two variants were excluded from Approach 2 based on this criterion, rs2149656 for WCadjBMI and rs2362267 for WHRadjBMI.

**Approaches.** Figure 1 outlines the four approaches that we used to identify novel SNPs. The left side of Fig. 1 focuses on the first hypothesis that examines the effect of SNPs on adiposity traits. Approach 1 considered a linear regression model that includes the SNP and SMK, thus adjusting for SMK (SNPadjSMK). Summary SNPadjSMK results were obtained from the SMK-adjusted meta-analysis. Approach 2 used summary SMK-stratified meta-analysis results[14] to consider the joint hypothesis that a genetic variant has main and/or interaction effects on outcomes as a 2 degree of freedom test (SNPjoint). For this approach, the null hypothesis was that there is no main and no interaction effect on the outcome. Thus, rejection of this hypothesis could be due to either a main effect or an interaction effect or to both.

The right side of Fig. 1 focuses on our second hypothesis, testing for interaction of a variant with SMK on adiposity traits as outcomes. Approach 3 used the SMK-stratified results to directly contrast the regression coefficients for a test of interaction (SNPint)[59]. Approach 4 used a screening strategy to evaluate interaction, whereby the SMK-adjusted main effect results (Approach 1) were screened for variants significant at the $P < 5 \times 10^{-8}$ level. These variants were then carried forward for a test of interaction, comparing the SMK-stratified specific regression coefficients in the second step (SNPscreen).

In Approaches 1–3 variants significant at $P < 5 \times 10^{-8}$ were considered GWS. In Approach 4 (SNPscreen) variants for which the $P$ value of the test of interaction is less than 0.05 divided by the number of variants carried forward were considered significant for interaction. We performed analytical power computations to demonstrate the usefulness and characteristic of the two interaction Approaches.

**Locuszoom plots.** Regional association plots were generated for novel loci using the program Locuszoom (http://locuszoom.sph.umich.edu/) . For each plot, LD was calculated using a multiethnic sample of the 1000 Genomes Phase I reference panels[62], including EUR, AFR, EAS and AMR. Previous SNP-trait associations highlighted within the plots include traits of interest (for example, cardiometabolic, addiction, behaviour and anthropometrics) found in the NHGRI-EMI GWAS Catalog and supplemented with recent GWAS studies from the literature[1,2,51,60].

**Conditional analyses.** To determine if multiple association signals were present within a single locus, we used GCTA[15] to perform approximate joint conditional analyses on the SNPadjSMK and SMK- stratified data. The following criteria were used to select candidate loci for conditional analyses: nearby SNP ($\pm 500$ kb) with an $R^2 > 0.4$ and an association $P < 1E^{-5}$ for any of our primary analyses. GCTA uses associations from our meta-analyses and LD estimates from reference data sets containing individual-level genotypic data to perform the conditional analyses. To calculate the LD structure, we used two U.S. cohorts, the Atherosclerosis Risk in Communities (ARIC) study consisting of 9,713 individuals of European descent and 580 individuals of African American descent, and the Framingham Heart Study (FramHS) consisting of 8,481 individuals of European ancestry, both studies imputed to HapMap r22. However, because our primary analyses were conducted in multiple ancestries, each study supplemented the genetic data using HapMap reference populations so that the final reference panel was composed of about 1–3% Asians (CHB + JPT) and 4–6% Africans (YRI for the FramHS) for the entire reference sample. We extracted each 1 MB region surrounding our candidate SNPs, performed joint approximate conditional analyses, and then repeated the steps for the appropriate Approach to identify additional association signals.

Many of the SNPs identified in the current analyses were nearby SNPs previously associated with related anthropometric and obesity traits (for example, height, visceral adipose tissue). For all lead SNPs near a SNP previously associated with these traits, GCTA was also used to perform approximate conditional analyses on the SNPadjSMK and SMK-stratified data in order to determine if the loci identified here are independent of the previously identified SNP-trait associations.

**Power and type I error.** In order to illustrate the validity of the approaches with regards to type 1 error, we conducted simulations. For two MAF, we assumed standardized stratum-specific outcomes for 50,000 smokers and 180,000 nonsmokers and generated 10,000 simulated stratum-specific effect sizes under the stratum-specific null hypotheses of 'no stratum-specific effects'. We applied the four approaches to the simulated stratum-specific association results and inferred type 1 error of each approach by visually examining QQ plots and by calculating type 1 error rates. The type 1 error rates shown reflect the proportion of nominally significant simulation results for the respective approach. Analytical power calculations to identify effects for various combinations of SMK- and NonSMK-specific effects by the Approaches 1–4 again assumed 50,000 smokers and 180,000 nonsmokers. We first assumed three different fixed effect estimates in smokers that were small ($R^2_{SMK} = 0.01\%$, similar to the realistic NUDT3 effect on

BMI), medium ($R^2_{SMK} = 0.07\%$, similar to the realistic BDNF effect on BMI) or large ($R^2_{SMK} = 0.34\%$, similar to the realistic FTO effect on BMI) genetic effects, and varied the effect in nonsmokers. Second, we assumed fixed (small, medium and large) effects in nonsmokers and varied the effect in smokers.

**Biological summaries.** To identify genes that may be implicated in the association between our lead SNPs (Tables 1–3) and BMI, WHRadjBMI and WCadjBMI, and to shed light on the complex relationship between genetic variants, SMK and adiposity, we performed in-depth literature searches on nearby candidate genes. Snipper v1.2 (http://csg.sph.umich.edu/boehnke/snipper/) was used to identify any genes and cis- or trans-eQTLs within 500 kb of our lead SNPs. All genes identified by Snipper were manually curated and examined for evidence of relationship with smoking and/or adiposity. To explore any potential regulatory or function role of the association regions, loci were also examined using several online bioinformatic tools/databases, including HaploReg v4.1 (ref. 63), UCSC Genome Browser (http://genome.ucsc.edu/), GTEx Portal (http://www.gtexportal.org), and RegulomeDB[64].

**eQTL analyses.** We used two approaches to systematically explore the role of novel loci in regulating gene expression. First, to gain a general overview of the regulatory role of newly identified GWAS regions, we conducted an eQTL lookup using >50 eQTL studies[65], with specific citations for >100 data sets included in the current query for blood cell related eQTL studies and relevant non-blood cell tissue eQTLs (for example, adipose and brain tissues). Additional eQTL data was integrated from online sources including ScanDB, the Broad Institute GTEx Portal, and the Pritchard Lab (eqtl.uchicago.edu). Additional details on the methods, including study references can be found in Supplementary Note 3. Only significant cis-eQTLS in high LD with our novel lead SNPs ($r^2 > 0.9$, calculated in the CEU + YRI + CHB + JPT 1000 Genomes reference panel), or proxy SNPs, were retained for consideration.

Second, since public databases with eQTL data do not have information available on current smoking status, we also conducted a cis-eQTL association analysis using expression results derived from fasting peripheral whole blood using the Human Exon 1.0 ST Array (Affymetrix, Inc., Santa Clara, CA). The raw expression data were quantile-normalized, log2 transformed, followed by summarization using Robust Multi-array Average[66] and further adjusted for technical covariates, including the first principal component of the expression data, batch effect, the all-probeset-mean residual, blood cell counts, and cohort membership. We evaluated all transcripts $\pm 1$ Mb around each novel variant in the Framingham Heart Study while accounting for current smoking status, using the following four approaches similar to those used in our primary analyses of our traits: (1) eQTL adjusted for SMK, (2) eQTL stratified by SMK, (3) eQTL × SMK interaction and (4) joint main + eQTLxSMK interaction). Significance level was evaluated by FDR < 5% per eQTL analysis and across all loci identified for that model in the primary meta-analysis. Additional details can be found in Supplementary Note 3.

**Variance-explained estimates.** We estimated the phenotypic variance in smokers and nonsmokers explained by the association signals. For each associated region, we selected subsets of SNPs within 500 kb of our lead SNPs and based on varying P value thresholds (ranging from $1 \times 10^{-8}$ to 0.1) from Approach 1 (SNPadjSMK model). First, each subset of SNPs was clumped into independent regions to identify the lead SNP for each region. The variance explained by each subset of SNPs in the SMK and nonSMK strata was estimated by summing the variance explained by the individual lead SNPs. Then, we tested for the significance of the differences across the two strata assuming that the weighted sum of chi-squared distributed variables tend to a Gaussian distribution ensured by Lyapunov's central limit theorem[67,68].

**Smoking behaviour lookups.** In order to determine if any of the loci identified in the current study are associated with smoking behaviour, we conducted a look-up of all lead SNPs from novel loci and Approach 3 in existing GWAS of smoking behaviour[3]. The analysis consists of phasing study-specific GWAS samples contributing to the smoking behaviour meta-analysis, imputation, association testing and meta-analysis. To ensure that all SNPs of interest were available in the smoking GWAS, the program SHAPEIT2 (ref. 69) was used to phase a region 500Kb either side of each lead SNP, and imputation was carried out using IMPUTE2 (ref. 70) with the 1000 Genomes Phase 3 data set as a reference panel.

Each region was analysed for three smoking related phenotypes: (i) Ever vs Never smokers, (ii) Current vs Non-current smokers and (iii) a categorical measure of smoking quantity[48]. The smoking quantity levels were 0 (defined as 1-10 cigarettes per day [CPD]), 1 (11-20 CPD), 2 (21-30 CPD) and 3 (31 or more CPD). Each increment represents an increase in smoking quantity of 10 cigarettes per day. There were 10,058 Never smokers, 13,418 Ever smokers, 11,796 Non-current smokers, 6,966 Current smokers and 11,436 samples with the SQ phenotypes. SNPMETA[48] was used to perform an inverse-variance weighted fixed effects meta-analysis across cohorts at all SNPs in each region, and included a single GC correction. At each SNP, only those cohorts that had an imputation info score >0.5 were included in the meta-analysis.

**Main effects lookup in previous GIANT investigations.** To better understand why our novel variants remained undiscovered in previous investigations that did not take SMK into account, we also conducted a lookup of our novel variants in published GWAS results examining genetic main effects on BMI, WC, WCadjBMI, WHR, WHRadjBMI, and height[1,2,51].

**GWAS catalog lookups.** To further investigate the identified genetic variants in this study and to gain additional insight into their functionality and possible effects on related cardiometabolic traits, we searched for previous SNP–trait associations nearby our lead SNPs. PLINK was used to find all SNPs within 500 kb of any of our lead SNPs and calculate $r^2$ values using a combined ancestry (AMR, AFR, EUR, ASN) 1000 Genomes Phase 1 reference panel[62] to allow for LD calculation for SNPs on the Illumina Metabochip and to best estimate LD in our multiethnic GWAS. All SNPs within the specified regions were compared with the NHGRI-EBI (National Human Genome Research Institute, European Bioinformatics Institute) GWAS Catalog, version 1.0 (www.ebi.ac.uk/gwas)[49,50] for overlap, and distances between the two SNPs were calculated using STATA v14, for the chromosome and base pair positions based on human genome reference build 19. All previous associations within 500 kb and with an $R^2 > 0.5$ with our lead SNP were retained for further interrogation.

**Genetic risk score calculation.** We calculated several unweighted genetic risk scores (GRSs) for each individual in the population-based KORA-S3 and KORA-S4 studies (total $N = 3{,}457$). We compared GRSs limited to previously known lead SNPs (see Supplementary Data 7 for lists of previously known lead SNPs) with GRSs based on previously known and novel lead SNPs from the current study (see Supplementary Tables 1–4 for lists of novel lead SNPs). Risk scores were tested for association with the obesity trait using the following linear regression models: The unadjusted GRS model ($\text{TRAIT} = \beta_0 + \beta_1 \text{GRS}$), the adjusted GRS model ($\text{TRAIT} = \beta_0 + \beta_1 \text{GRS} + \beta_2 \text{SMK}$) and the GRSxSMK interaction model ($\text{TRAIT} = \beta_0 + \beta_1 \text{GRS} + \beta_2 \text{SMK} + \beta_3 \text{GRSxSMK}$). Additionally, we used an F statistic to test whether the residual sum of squares (RSS) for the full model including GRSxSMK interaction was significantly different from the reduced model.

**Data availability.** Summary statistics of all analyses are available at https://www.broadinstitute.org/collaboration/giant/.

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

## Acknowledgements

A full list of acknowledgements appears in the Supplementary Note 4. Co-author A.J.M.d.C. recently passed away while this work was in process. This work was performed under the auspices of the Genetic Investigation of ANthropometric Traits (GIANT) consortium. We acknowledge the Cohorts for Heart and Aging Research in Genomic Epidemiology (CHARGE) Consortium for encouraging CHARGE studies to participate in this effort and for the contributions of CHARGE members to the analyses conducted for this research. Funding for this study was provided by the Aase and Ejner Danielsens Foundation; Academy of Finland (41071, 77299, 102318, 110413, 117787, 121584, 123885, 124243, 124282, 126925, 129378, 134309, 286284); Accare Center for Child and Adolescent Psychiatry; Action on Hearing Loss (G51); Agence Nationale de la Recherche; Agency for Health Care Policy Research (HS06516); ALF/LUA research grant in Gothenburg; ALFEDIAM; ALK-Abelló A/S; Althingi; American Heart Association (13POST16500011); Amgen; Andrea and Charles Bronfman Philanthropies; Ardix Medical; Arthritis Research UK; Association Diabète Risque Vasculaire; Australian National Health and Medical Research Council (241944, 339462, 389875, 389891, 389892, 389927, 389938, 442915, 442981, 496739, 552485, 552498); Avera Institute; Bayer Diagnostics; Becton Dickinson; BHF (RG/14/5/30893); Boston Obesity Nutrition Research Center (DK46200); Bristol-Myers Squibb; British Heart Foundation (RG/10/12/28456, RG2008/08, RG2008/014, SP/04/002); Medical Research Council of Canada; Canadian Institutes for Health Research (FRCN-CCT-83028); Cancer Research UK; Cardionics; Cavadis B.V., Center for Medical Systems Biology; Center of Excellence in Genomics; CFI; CIHR; City of Kuopio; CNAMTS; Cohortes Santé TGIR; Contrat de Projets État-Région; Croatian Science Foundation (8875); Danish Agency for Science, Technology and Innovation; Danish Council for Independent Research (DFF-1333-00124, DFF-1331-00730B); County Council of Dalarna; Dalarna University; Danish Council for Strategic Research; Danish Diabetes Academy; Danish Medical Research Council; Department of Health, UK; Development Fund from the University of Tartu (SP1GVARENG); Diabetes Hilfs- und Forschungsfonds Deutschland; Diabetes UK; Diabetes Research and Wellness Foundation Fellowship; Donald W. Reynolds Foundation; Dr Robert Pfleger-Stiftung; Dutch Brain Foundation; Dutch Diabetes Research Foundation; Dutch Inter University Cardiology Institute; Dutch Kidney Foundation (E033); Dutch Ministry of Justice; the DynaHEALTH action No. 633595, Economic Structure Enhancing Fund of the Dutch Government; Else Kröner-Fresenius-Stiftung (2012_A147, P48/08//A11/08); Emil Aaltonen Foundation; Erasmus University Medical Center Rotterdam; Erasmus MC and Erasmus University Rotterdam; the Municipality of Rotterdam; Estonian Government (IUT20-60, IUT24-6); Estonian Research Roadmap through the Estonian Ministry of Education and Research (3.2.0304.11-0312); European Research Council (ERC Starting Grant and 323195:SZ-245 50371-GLUCOSEGENES-FP7-IDEAS-ERC); European Regional Development Fund; European Science Foundation (EU/QLRT-2001-01254); European Commission (018947, 018996, 201668, 223004, 230374, 279143, 284167, 305739, BBMRI-LPC-313010, HEALTH-2011.2.4.2-2-EU-MASCARA, HEALTH-2011-278913, HEALTH-2011-294713-EPLORE, HEALTH-F2-2008-201865-GEFOS, HEALTH-F2-2013-601456, HEALTH-F4-2007-201413, HEALTH-F4-2007-201550-HYPERGENES, HEALTH-F7-305507 HOMAGE, IMI/115006, LSHG-CT-2006-018947, LSHG-CT-2006-01947, LSHM-CT-2004-005272, LSHM-CT-2006-037697, LSHM-CT-2007-037273, QLG1-CT-2002-00896, QLG2-CT-2002-01254); Faculty of Biology and Medicine of Lausanne; Federal Ministry of Education and Research (01ZZ0103, 01ZZ0403, 01ZZ9603, 03IS2061A, 03ZIK012); Federal State of Mecklenburg-West Pomerania; Fédération Française de Cardiologie; Finnish Cultural Foundation; Finnish Diabetes Association; Finnish Foundation of Cardiovascular Research; Finnish Heart Association; Fondation Leducq; Food Standards Agency; Foundation for Strategic Research; French Ministry of Research; FRSQ; Genetic Association Information Network (GAIN) of the Foundation for the NIH; German Federal Ministry of Education and Research (BMBF, 01ER1206, 01ER1507); GlaxoSmithKline; Greek General Secretary of Research and Technology; Göteborg Medical Society; Health and Safety Executive; Healthcare NHS Trust; Healthway; Western Australia; Heart Foundation of Northern Sweden; Helmholtz Zentrum München—German Research Center for Environmental Health; Hjartavernd; Ingrid Thurings Foundation; INSERM; InterOmics (PB05 MIUR-CNR); INTERREG IV Oberrhein Program (A28); Inter-university Cardiology Institute of the Netherlands (ICIN, 09.001); Italian Ministry of Health (ICS110.1/RF97.71); Italian Ministry of Economy and Finance (FaReBio di Qualità); Marianne and Marcus Wallenberg Foundation; the Ministry of Health, Welfare and Sports, the Netherlands; J.D.E. and Catherine T, MacArthur Foundation Research Networks on Successful Midlife Development and Socioeconomic Status and Health; Juho Vainio Foundation; Juvenile Diabetes Research Foundation International; KfH Stiftung Präventivmedizin e.V.; King's College London; Knut and Alice Wallenberg Foundation; Kuopio University Hospital; Kuopio, Tampere and Turku University Hospital Medical Funds (X51001); La Fondation de France; Leenaards Foundation; Lilly; LMUinnovativ; Lundberg Foundation; Magnus Bergvall Foundation; MDEIE; Medical Research Council UK (G0000934, G0601966, G0700931, MC_U106179471, MC_UU_12019/1); MEKOS Laboratories; Merck Santé; Ministry for Health, Welfare and Sports, The Netherlands; Ministry of Cultural Affairs of Mecklenburg-West Pomerania; Ministry of Economic Affairs, The Netherlands; Ministry of Education and Culture of Finland (627;2004-2011); Ministry of Education, Culture and Science, The Netherlands; Ministry of Science, Education and Sport in the Republic of Croatia (108-1080315-0302); MRC centre for Causal Analyses in Translational Epidemiology; MRC Human Genetics Unit; MRC-GlaxoSmithKline pilot programme (G0701863); MSD Stipend Diabetes; National Institute for Health Research; Netherlands Brain Foundation (F2013(1)-28); Netherlands CardioVascular Research Initiative (CVON2011-19); Netherlands Genomics Initiative (050-060-810); Netherlands Heart Foundation (2001 D 032, NHS2010B280); Netherlands Organization for Scientific Research (NWO) and Netherlands Organisation for Health Research and Development (ZonMW) (56-464-14192, 60-60600-97-118, 100-001-004, 261-98-710, 400-05-717, 480-04-004, 480-05-003, 481-08-013, 904-61-090, 904-61-193, 911-11-025, 985-10-002, Addiction-31160008, BBMRI–NL 184.021.007, GB-MaGW 452-04-314, GB-MaGW 452-06-004, GB-MaGW 480-01-006, GB-MaGW 480-07-001, GB-MW 940-38-011, Middelgroot-911-09-032, NBIC/BioAssist/RK 2008.024, Spinozapremie 175.010.2003.005, 175.010.2007.006);

Neuroscience Campus Amsterdam; NHS Foundation Trust; National Institutes of Health (1RC2MH089951, 1Z01HG000024, 24152, 263MD9164, 263MD821336, 2R01LM010098, 32100-2, 32122, 32108, 5K99HL130580-02, AA07535, AA10248, AA11998, AA13320, AA13321, AA13326, AA14041, AA17688, AG13196, CA047988, DA12854, DK56350, DK063491, DK078150, DK091718, DK100383, DK078616, ES10126, HG004790, HHSN268200625226C, HHSN268200800007C, HHSN268201200036C, HHSN268201500001I, HHSN268201100046C, HHSN268201100001C, HHSN268201100002C, HHSN268201100003C, HHSN268201100004C, HHSN271201100004C, HL043851, HL45670, HL080467, HL085144, HL087660, HL054457, HL119443, HL118305, HL071981, HL034594, HL126024, HL130114, KL2TR001109, MH66206, MH081802, N01AG12100, N01HC55015, N01HC55016, N01C55018, N01HC55019, N01HC55020, N01HC55021, N01HC55022, N01HC85079, N01HC85080, N01HC85081, N01HC85082, N01HC85083, N01HC85086, N01HC95159, N01HC95160, N01HC95161, N01HC95162, N01HC95163, N01HC95164, N01HC95165, N01HC95166, N01HC95167, N01HC95168, N01HC95169, N01HG65403, N01WH22110, N02HL6-4278, N01-HC-25195, P01CA33619, R01HD057194, R01HD057194, R01AG023629, R01CA63, R01D004215701A, R01DK075787, R01DK062370, R01DK072193, R01DK075787, R01DK089256, R01HL53353, R01HL59367, R01HL086694, R01HL087641, R01HL087652, R01HL103612, R01HL105756, R01HL117078, R01HL120393, R03 AG046389, R37CA54281, RC2AG036495, RC4AG039029, RPPG040710371, RR20649, TW008288, TW055596, U01AG009740, U01CA98758, U01CA136792, U01DK062418, U01HG004402, U01HG004802, U01HG007376, U01HL080295, UL1RR025005, UL1TR000040, UL1TR000124, UL1TR001079, 2T32HL007055-36, T32GM074905, HG002651, HL084729, N01-HC-25195, UM1CA182913); NIH, National Institute on Aging (Intramural funding, NO1-AG-1-2109); Northern Netherlands Collaboration of Provinces; Novartis Pharma; Novo Nordisk; Novo Nordisk Foundation; Nutricia Research Foundation (2016-T1); ONIVINS; Parnassia Bavo group; Pierre Fabre; Province of Groningen; Päivikki and Sakari Sohlberg Foundation; Påhlssons Foundation; Paavo Nurmi Foundation; Radboud Medical Center Nijmegen; Research Centre for Prevention and Health, the Capital Region of Denmark; the Research Institute for Diseases in the Elderly; Research into Ageing; Robert Dawson Evans Endowment of the Department of Medicine at Boston University School of Medicine and Boston Medical Center; Roche; Royal Society; Russian Foundation for Basic Research (NWO-RFBR 047.017.043); Rutgers University Cell and DNA Repository (NIMH U24 MH068457-06); Sanofi-Aventis; Scottish Government Health Directorates, Chief Scientist Office (CZD/16/6); Siemens Healthcare; Social Insurance Institution of Finland (4/26/2010); Social Ministry of the Federal State of Mecklenburg-West Pomerania; Société Francophone du 358 Diabète; State of Bavaria; Stiftelsen för Gamla Tjänarinnor; Stockholm County Council (560183, 592229); Strategic Cardiovascular and Diabetes Programmes of Karolinska Institutet and Stockholm County Council; Stroke Association; Swedish Diabetes Association; Swedish Diabetes Foundation (2013-024); Swedish Foundation for Strategic Research; Swedish Heart-Lung Foundation (20120197, 20150711); Swedish Research Council (0593, 8691, 2012-1397, 2012-1727, and 2012-2215); Swedish Society for Medical Research; Swiss Institute of Bioinformatics; Swiss National Science Foundation (3100AO-116323/1, 31003A-143914, 33CSCO-122661, 33CS30-139468, 33CS30-148401, 51RTP0_151019); Tampere Tuberculosis Foundation; Technology Foundation STW (11679); The Fonds voor Wetenschappelijk Onderzoek Vlaanderen, Ministry of the Flemish Community (G.0880.13, G.0881.13); The Great Wine Estates of the Margaret River Region of Western Australia; Timber Merchant Vilhelm Bangs Foundation; Topcon; Tore Nilsson Foundation; Torsten and Ragnar Söderberg's Foundation; United States – Israel Binational Science Foundation (Grant 2011036); Umeå University; University Hospital of Regensburg; University of Groningen; University Medical Center Groningen; University of Michigan; University of Utrecht; Uppsala Multidisciplinary Center for Advanced Computational Science (UPPMAX) (b2011036); Velux Foundation; VU University's Institute for Health and Care Research; Västra Götaland Foundation; Wellcome Trust (068545, 076113, 079895, 084723, 088869, WT064890, WT086596, WT098017, WT090532, WT098051, 098381); Wissenschaftsoffensive TMO; Yrjö Jahnsson Foundation; and Åke Wiberg Foundation. The views expressed in this manuscript are those of the authors and do not necessarily represent the views of the National Heart, Lung, and Blood Institute (NHLBI); the National Institutes of Health (NIH); or the U.S. Department of Health and Human Services.

## Author contributions

L.A.C., K.E.N., I.B.B., T.O.K., R.J.F.L. and C.T.L. supervised this project together. L.A.C., K.E.N., I.B.B., T.O.K., T.W.W., R.J.F.L. and A.E.J. conceived and designed the study. L.A.C., R.J.F.L., A.E.J. and T.O.K. coordinated the collection of genome-wide association and interaction study results from the participating studies. The association and interaction results were contributed by S.W.v.d.L., M.A.Si., S.H., G.J.d.B., H.M.D.R. and G.P. (AtheroExpress); A.V.S., T.B.H., G.E., L.J.L. and V.G. (AGES study); K.E.N., M.Gr., A.E.J., K.Y., E.Boe. and P.G.L. (ARIC study); J.B.W., N.G.M., R.P.S.M., P.A.F.M., A.C.H. and G.W.M. (AUSTWIN study); D.P.S. (B58C study); G.C., L.J.P., J.oH., A.W.M., A.L.J. and J.Be. (BHS study); C.Sch., T.A., E.Bot. and R.J.F.L. (BioMe); T.T., D.He., L.F. (BLSA); B.M., T.M.B., K.D.T., S.C. and B.M.P. (CHS); Y.W., N.R.L., L.S.A. and K.L.M. (CLHNS study); Z.K., P.M.V., T.C., S.Be., G.Wa. and P.V. (COLAUS study); J.Mart., I.R. and C.H. (Croatia-Korcula study); V.V., I.K. and O.Po. (Croatia-Vis study); L.Y., A.B., D.T., S.Lo., B.B. and P.F. (DESIR study); R.Rau., T.A.L., P.K., M.Ha., K.Sa. and R.M. (DR's EXTRA

study); K.F., N.P., T.E. and A.Me. (EGCUT study); J.L., R.A.S., C.L. and N.J.W. (Ely study); C.L., J.L., R.A.S. and N.J.W. (EPIC); J.H.Z., R.L., R.A.S. and N.J.W. (EPIC-Norfolk study); N.A., M.C.Z. and C.Mv.D. (ERF study); I.B.B., M.F.F., J.C. and L.B. (Family Heart Study); J.L., R.A.S., C.L., R.J.F.L. and N.J.W. (Fenland study); F.X., J.W., J.S.N., V.A.F., N.L.H.C., C.T.L., C.S.F. and L.A.C. (FramHS); M.Bo., F.S.C., K.L.M. and R.N.B. (FUSION study); J.T., L.K., C.Sa. and H.A.K. (FUSION2 study); M.Go., B.K.K. and C.A.B. (Gendian); D.J.P., J.E.H., L.J.H., S.P., C.H. and B.H.S. (Generation Scotland); L.F.B., S.L.R.K., M.A.J. and P.A.P. (GENOA study); S.Ah., F.R., I.B., G.Ha. and P.W.F. (GLACIER study); J.E., C.O., J.O.J., M.Lor., A.E. and L.V. (GOOD study); T.S.A., T.Ha. and T.I.A.S. (GOYA study); B.O.T., C.A.M., S.L.V., T.F., J.N.H. and R.S.C. (GxE); M.Hol., M.N.H., C.P., A.L. and H.Ve. (Health06 study); Y.J.S., T.Ri., T.Ra., M.A.Sa., D.C.R. and C.B. (HERITAGE Family Study); J.A.Sm., J.D.F., S.L.R.K., W.Zhao. and D.R.W. (HRS study); A.U.J., K.K., O.L.H., L.L.B., A.J.W. and K.H. (HUNT2 study); M.C., D.Br., S.Lu., N.Gl., J.A.St. and D.C. (HYPERGENES); R.J.S., B.S., K.G., U.dF., A.H., E.T. and D.Ba. (IMPROVE); T.T., D.He. and S.Ba. (InCHIANTI study); J.M.J., M.E.J., N.Gr. and O.Pe. (Inter99 study); T.W.W., I.M.H., M.E.Z., M.M.N., M.O., A.L.D., H.G., M.W., R.Raw., B.T., A.P. and K.St. (KORA S3 and S4 studies); J.V.vV.O., J.M.V., S.Sch., M.A.Sw. and B.H.R.W. (Lifelines); W.Zhang., M.Lo., U.A., S.Af., J.C.C. and J.S.K. (LOLIPOP study); M.E.K., G.E.D., T.B.G., G.S., Ji.H. and W.M. (LURIC study); U.L., C.A.H., L.Le.M., S.Bu. and L.H. (MEC study); A.Man., L.J.R.T. and Y.dI.C. (MESA study); M.La., J.K., A.J.S., H.M.S., P.S.C. and N.N. (METSIM study); M.Ka., D.M. and C.O. (MrOS); M.Hor., M.R.J. and M.I.M. (NFBC66 study); L.Q., T.Hu., Q.Q. and D.J.H. (NHS study); D.K., K.K.O., J.L. and A.W. (NSHD study); J.M.V., G.Wi., G.L., J.J.H., E.J.C.dG. and D.I.B. (NTR study); P.N., A.F.W., N.D.H., S.W., H.C. and J.F.W. (ORCADES study); A.Mah., C.M.L., E.I., L.L. and A.P.M. (PIVUS); N.V., S.J.L.B. and P.vdH. (Prevend); S.T., D.J.S., B.M.B., A.J.Md.C., I.F., R.W., P.E.S., N.S. and J.W.J. (PROSPER); L.P., M.C.V., J.E.C., J.Bl. and C.B. (QFS study); N.D., M.C.Z., A.G.U. and H.T. (RS1/RS2/RS3 study); J.Br., S.Sa., D.S., G.R.A. and F.C. (SardiNIA study); R.J.S., B.S., B.G., K.L., A.H. and U.dF. (SCARFSHEEP); A.T., R.B., G.Ho., M.N., H.Vö. and H.J.G. (SHIP study); B.O.T., C.A.M., S.L.V., T.F., J.N.H. and R.S.C. (SPT); S.K., G.K., G.D. and P.D. (THISEAS); P.J.vd.M., I.M.N., H.S., A.J.O., C.A.H. and M.Br. (TRAILS study); M.Ma., C.M. and T.D.S. (TwinsUK study); A.Y.C., L.M.R., P.M.R. and D.I.C. (WGHS study); N.Z., S.R., J.G., C.K. and U.P. (WHI study); M.Ku., C.L., J.L. and M.Ki. (Whitehall study); L.P.L., N.H.K., M.J., M.Kä., O.T.R. and T.L. (YFS study). T.W.W., M.Gr., K.Y., J.C., D.Ha., J.S.N., T.S.A., N.L.H.C., F.R., L.X., Q.Q., J.W. and A.E.J. cleaned and quality checked the association and interaction results from the participating studies. T.W.W., K.Y., V.A.F., X.D., J.C., D.Ha., J.S.N., T.S.A., N.L.H.C., L.X. and A.E.J. performed the meta-analyses. A.Y.C., A.E.J., L.L.B., M.F.F., T.O.K. and L.A.C. collected the Supplementary Information from the participating studies. A.E.J., M.Gr., M.F.F., K.Y. and V.A.F. organized the Supplementary Tables. D.Ha., T.W.W. and A.E.J. provided look-up information from the GWAS meta-analysis of BMI, WAISTadjBMI and WHRadjBMI. JMarc provided lookup information from Smoking GWAS meta-analysis. A.E.J. performed the look-up in the NHGRI-EBI GWAS Catalog. M.Gr., X.D., A.E.J. and Z.K. performed the analyses for variance explained by common variants in the SMK and nonSMK groups. M.F.F., K.Y., C.T.L., X.D., L.B. and A.E.J. reviewed the literature for the identified loci. A.E.J., K.Y., V.A.F. and M.G. performed approximate conditional analyses. T.W.W. conducted power and type 1 error simulations. A.E.J. produced heatmap and forest plots. J.D.E. and A.D.J. carried out the lookups for Expression Quantitative Trait loci. J.P., E.L. and C.T.L. conducted eQTL analyses in the Framingham Heart Study. J.Ty. and T.Fr. conducted validation analyses in UKBB. A.E.J., M.Gr. and K.Y. conducted meta-analyses of GIANT and UKBB results. A.E.J., T.W.W., M.F.F., M.Gr., K.Y., V.A.F., X.D., L.B., J.Marc., T.O.K., C.T.L., J.S.N., R.J.F.L., K.E.N. and L.A.C. wrote the manuscript.

## Additional information

**Competing interests:** B.M.P. serves on the DSMB for a clinical trial funded by the device manufacturer (Zoll LifeCor) and on the Steering Committee of the Yale Open Data Access Project funded by Johnson & Johnson. The remaining authors declare no competing financial interests.

Anne E. Justice[1,*], Thomas W. Winkler[2,*], Mary F. Feitosa[3,*], Misa Graff[1,*], Virginia A. Fisher[4,*], Kristin Young[1,*], Llilda Barata[3,*], Xuan Deng[4], Jacek Czajkowski[3], David Hadley[5,6], Julius S. Ngwa[4,7], Tarunveer S. Ahluwalia[8,9], Audrey Y. Chu[10,11], Nancy L. Heard-Costa[10,12], Elise Lim[4], Jeremiah Perez[4], John D. Eicher[13], Zoltán Kutalik[14,15,16], Luting Xue[4], Anubha Mahajan[17], Frida Renström[18,19], Joseph Wu[4], Qibin Qi[20], Shafqat Ahmad[11,19,21], Tamuno Alfred[22,23], Najaf Amin[24], Lawrence F. Bielak[25], Amelie Bonnefond[26], Jennifer Bragg[27,28], Gemma Cadby[29], Martina Chittani[30], Scott Coggeshall[31], Tanguy Corre[14,15,16], Nese Direk[32,33], Joel Eriksson[34], Krista Fischer[35], Mathias Gorski[2,36], Marie Neergaard Harder[8], Momoko Horikoshi[17,37], Tao Huang[21,38], Jennifer E. Huffman[13,39], Anne U. Jackson[28], Johanne Marie Justesen[8], Stavroula Kanoni[40], Leena Kinnunen[41], Marcus E. Kleber[42], Pirjo Komulainen[43], Meena Kumari[44,45], Unhee Lim[46], Jian'an Luan[47], Leo-Pekka Lyytikäinen[48,49], Massimo Mangino[50,51], Ani Manichaikul[52], Jonathan Marten[39], Rita P.S. Middelberg[53], Martina Müller-Nurasyid[54,55,56], Pau Navarro[39], Louis Pérusse[57,58], Natalia Pervjakova[35,59], Cinzia Sarti[60], Albert Vernon Smith[61,62], Jennifer A. Smith[25], Alena Stančáková[63], Rona J. Strawbridge[64,65], Heather M. Stringham[28], Yun Ju Sung[66], Toshiko Tanaka[67], Alexander Teumer[68], Stella Trompet[69,70], Sander W. van der Laan[71], Peter J. van der Most[72], Jana V. Van Vliet-Ostaptchouk[73], Sailaja L. Vedantam[74,75], Niek Verweij[76], Jacqueline M. Vink[77,78], Veronique Vitart[39], Ying Wu[79], Loic Yengo[26], Weihua Zhang[80,81], Jing Hua Zhao[47], Martina E. Zimmermann[2], Niha Zubair[82], Gonçalo R. Abecasis[28], Linda S. Adair[83], Saima Afaq[80,81], Uzma Afzal[80,81], Stephan J.L. Bakker[84], Traci M. Bartz[31,85], John Beilby[86,87,88], Richard N. Bergman[89], Sven Bergmann[15,16], Reiner Biffar[90], John Blangero[91], Eric Boerwinkle[92], Lori L. Bonnycastle[93], Erwin Bottinger[22,94], Daniele Braga[30], Brendan M. Buckley[95], Steve Buyske[96,97], Harry Campbell[98], John C. Chambers[81,80,99], Francis S. Collins[93], Joanne E. Curran[91], Gert J. de Borst[100], Anton J.M. de Craen[70,‡], Eco J.C. de Geus[77,101], George Dedoussis[102], Graciela E. Delgado[42], Hester M. den Ruijter[71], Gudny Eiriksdottir[61], Anna L. Eriksson[34], Tõnu Esko[35,74,75], Jessica D. Faul[103], Ian Ford[104], Terrence Forrester[105], Karl Gertow[64,65], Bruna Gigante[106], Nicola Glorioso[107], Jian Gong[82], Harald Grallert[108,109,110], Tanja B. Grammer[42], Niels Grarup[8], Saskia Haitjema[71], Göran Hallmans[111], Anders Hamsten[64,65], Torben Hansen[8], Tamara B. Harris[112], Catharina A. Hartman[113], Maija Hassinen[43], Nicholas D. Hastie[39], Andrew C. Heath[114], Dena Hernandez[115], Lucia Hindorff[116], Lynne J. Hocking[117,118], Mette Hollensted[8], Oddgeir L. Holmen[119], Georg Homuth[120], Jouke Jan Hottenga[77], Jie Huang[121], Joseph Hung[122,123], Nina Hutri-Kähönen[124,125], Erik Ingelsson[126,127,128], Alan L. James[86,122,129], John-Olov Jansson[130], Marjo-Riitta Jarvelin[131,132,133,134], Min A. Jhun[25], Marit E. Jørgensen[9], Markus Juonala[135,136], Mika Kähönen[137,138], Magnus Karlsson[139], Heikki A. Koistinen[41,140,141], Ivana Kolcic[142], Genovefa Kolovou[143], Charles Kooperberg[82], Bernhard K. Krämer[42], Johanna Kuusisto[144], Kirsti Kvaløy[145], Timo A. Lakka[43,146], Claudia Langenberg[47], Lenore J. Launer[112], Karin Leander[106], Nanette R. Lee[147,148], Lars Lind[149], Cecilia M. Lindgren[17,150], Allan Linneberg[151,152,153], Stephane Lobbens[26], Marie Loh[80,154], Mattias Lorentzon[34], Robert Luben[155], Gitta Lubke[156], Anja Ludolph-Donislawski[54,157], Sara Lupoli[30], Pamela A.F. Madden[114], Reija Männikkö[43], Pedro Marques-Vidal[158], Nicholas G. Martin[53], Colin A. McKenzie[105], Barbara McKnight[31,85,159], Dan Mellström[34], Cristina Menni[50], Grant W. Montgomery[160], A.W. (Bill) Musk[86,161,162], Narisu Narisu[93], Matthias Nauck[163], Ilja M. Nolte[72], Albertine J. Oldehinkel[113], Matthias Olden[2], Ken K. Ong[47], Sandosh Padmanabhan[118,164], Patricia A. Peyser[25], Charlotta Pisinger[165,166], David J. Porteous[118,167], Olli T. Raitakari[168,169], Tuomo Rankinen[170], D.C. Rao[66,114,171], Laura J. Rasmussen-Torvik[172], Rajesh Rawal[108,109], Treva Rice[66,114], Paul M. Ridker[11,173], Lynda M. Rose[11], Stephanie A. Bien[82], Igor Rudan[98], Serena Sanna[174], Mark A. Sarzynski[170], Naveed Sattar[175], Kai Savonen[43], David Schlessinger[176], Salome Scholtens[72], Claudia Schurmann[22,23], Robert A. Scott[47], Bengt Sennblad[64,65,177], Marten A. Siemelink[71], Günther Silbernagel[178], P. Eline Slagboom[179], Harold Snieder[72], Jan A. Staessen[180,181], David J. Stott[182], Morris A. Swertz[183], Amy J. Swift[93], Kent D. Taylor[184,185], Bamidele O. Tayo[186], Barbara Thorand[109,110], Dorothee Thuillier[26],

Jaakko Tuomilehto[187,188,190], Andre G. Uitterlinden[32,191], Liesbeth Vandenput[34], Marie-Claude Vohl[58,192], Henry Völzke[68], Judith M. Vonk[72], Gérard Waeber[158], Melanie Waldenberger[108,109], R.G.J. Westendorp[193], Sarah Wild[98], Gonneke Willemsen[77], Bruce H.R. Wolffenbuttel[73], Andrew Wong[194], Alan F. Wright[39], Wei Zhao[25], M. Carola Zillikens[191], Damiano Baldassarre[195,196], Beverley Balkau[197], Stefania Bandinelli[198], Carsten A. Böger[36], Dorret I. Boomsma[77], Claude Bouchard[170], Marcel Bruinenberg[199], Daniel I. Chasman[11,200], Yii-Der Ida Chen[201], Peter S. Chines[93], Richard S. Cooper[186], Francesco Cucca[174,202], Daniele Cusi[203], Ulf de Faire[106], Luigi Ferrucci[67], Paul W. Franks[19,21,204], Philippe Froguel[26,205], Penny Gordon-Larsen[83,206], Hans-Jörgen Grabe[207,208], Vilmundur Gudnason[61,62], Christopher A. Haiman[209], Caroline Hayward[39,118], Kristian Hveem[145], Andrew D. Johnson[13], J. Wouter Jukema[69,210,211], Sharon L.R. Kardia[25], Mika Kivimaki[45], Jaspal S. Kooner[81,99,212], Diana Kuh[194], Markku Laakso[144], Terho Lehtimäki[48,49], Loic Le Marchand[46], Winfried März[213,214], Mark I. McCarthy[37,17,215], Andres Metspalu[35], Andrew P. Morris[17,216], Claes Ohlsson[34], Lyle J. Palmer[217], Gerard Pasterkamp[71,218], Oluf Pedersen[8], Annette Peters[109,110], Ulrike Peters[82], Ozren Polasek[98,142], Bruce M. Psaty[219,220,221], Lu Qi[21,222], Rainer Rauramaa[43,223], Blair H. Smith[118,224], Thorkild I.A. Sørensen[8,225,226], Konstantin Strauch[54,157], Henning Tiemeier[227], Elena Tremoli[195,196], Pim van der Harst[76,183,228], Henrik Vestergaard[8,9], Peter Vollenweider[158], Nicholas J. Wareham[47], David R. Weir[103], John B. Whitfield[53], James F. Wilson[39,229], Jessica Tyrrell[230,231], Timothy M. Frayling[232], Inês Barroso[233,234,235], Michael Boehnke[28], Panagiotis Deloukas[40,233,236], Caroline S. Fox[10], Joel N. Hirschhorn[74,75,237], David J. Hunter[21,75,238,239], Tim D. Spector[50], David P. Strachan[5,240], Cornelia M. van Duijn[24,241,242], Iris M. Heid[2,243], Karen L. Mohlke[79], Jonathan Marchini[244], Ruth J.F. Loos[22,23,47,245,246,**], Tuomas O. Kilpeläinen[8,47,247,**], Ching-Ti Liu[4,**], Ingrid B. Borecki[3,**], Kari E. North[1,**] & L. Adrienne Cupples[4,10,**]

[1] Department of Epidemiology, University of North Carolina, Chapel Hill, North Carolina 27599, USA. [2] Department of Genetic Epidemiology, Institute of Epidemiology and Preventive Medicine, University of Regensburg, D-93053 Regensburg, Germany. [3] Division of Statistical Genomics, Department of Genetics, Washington University School of Medicine; St. Louis, Missouri 63108, USA. [4] Department of Biostatistics, Boston University School of Public Health, Boston, Massachusetts 02118, USA. [5] Population Health Research Institute, St. George's, University of London, London SW17 0RE, UK. [6] TransMed Systems, Inc., Cupertino, California 95014, USA. [7] Department of Biostatistics, Johns Hopkins Bloomberg School of Public Health, Baltimore, Maryland, USA. [8] The Novo Nordisk Foundation Center for Basic Metabolic Research, Section of Metabolic Genetics, Faculty of Health and Medical Sciences, University of Copenhagen, Copenhagen, Denmark. [9] Steno Diabetes Center, Gentofte, Denmark. [10] NHLBI Framingham Heart Study, Framingham, Massachusetts 01702, USA. [11] Division of Preventive Medicine, Brigham and Women's Hospital and Harvard Medical School, Boston, Massachusetts, USA. [12] Department of Neurology, Boston University School of Medicine, Boston, Massachusetts 02118, USA. [13] Population Sciences Branch, National Heart, Lung, and Blood Institute, National Institutes of Health, The Framingham Heart Study, Framingham, Massachusetts, USA. [14] Institute of Social and Preventive Medicine (IUMSP), Centre Hospitalier Universitaire Vaudois (CHUV), Lausanne, Switzerland. [15] Department of Computational Biology, University of Lausanne, Lausanne, Switzerland. [16] Swiss instititute of Bioinformatics, 1015 Lausanne, Switzerland. [17] Wellcome Trust Centre for Human Genetics, University of Oxford, Oxford OX3 7BN, UK. [18] Department of Biobank Research, Umeå University, Umeå, Sweden. [19] Department of Clinical Sciences, Genetic and Molecular Epidemiology Unit, Lund University, SE-205 02 Malmö, Sweden. [20] Department of Epidemiology and Population Health, Albert Einstein College of Medicine, Bronx, New York, USA. [21] Department of Nutrition, Harvard T.H. Chan School of Public Health, Boston, Massachusetts 02115, USA. [22] The Charles Bronfman Institute for Personalized Medicine, Icahn School of Medicine at Mount Sinai, New York, USA. [23] The Genetics of Obesity and Related Metabolic Traits Program, Icahn School of Medicine at Mount Sinai, New York, USA. [24] Genetic Epidemiology Unit, Department of Epidemiology, Erasmus University Medical Center, Rotterdam 3015GE, The Netherlands. [25] Department of Epidemiology, School of Public Health, University of Michigan, Ann Arbor, Michigan, USA. [26] University of Lille, CNRS, Institut Pasteur of Lille, UMR 8199 - EGID, Lille, France. [27] Internal Medicine - Nephrology, University of Michigan, Ann Arbor, Michigan, USA. [28] Department of Biostatistics and Center for Statistical Genetics, University of Michigan, Ann Arbor, Michigan 48109, USA. [29] Centre for Genetic Origins of Health and Disease, University of Western Australia, Crawley 6009, Australia. [30] Department of Health Sciences, University of Milan,Via A. Di Rudiní, 8 20142, Milano, Italy. [31] Department of Biostatistics, University of Washington, Seattle, Washington 98195, USA. [32] Department of Epidemiology, Erasmus Medical Center, Rotterdam, The Netherlands. [33] Department of Psychiatry, Dokuz Eylul University, Izmir, Turkey. [34] Centre for Bone and Arthritis Research, Department of Internal Medicine and Clinical Nutrition, Institute of Medicine, Sahlgrenska Academy at the University of Gothenburg, Gothenburg, Sweden. [35] Estonian Genome Center, University of Tartu, Tartu 51010, Estonia. [36] Department of Nephrology, University Hospital Regensburg, Regensburg, Germany. [37] Oxford Centre for Diabetes, Endocrinology and Metabolism, University of Oxford, Churchill Hospital, Oxford OX3 7LJ, UK. [38] Epidemiology Domain, Saw Swee Hock School of Public Health, National University of Singapore, Singapore 117549, Singapore. [39] MRC Human Genetics Unit, Institute of Genetics and Molecular Medicine, University of Edinburgh, Edinburgh, Scotland. [40] William Harvey Research Institute, Barts and The London School of Medicine and Dentistry, Queen Mary University of London, London, UK. [41] Department of Health, National Institute for Health and Welfare, Helsinki FI-00271, Finland. [42] Vth Department of Medicine, Medical Faculty Mannheim, Heidelberg University, Mannheim, Germany. [43] Kuopio Research Institute of Exercise Medicine, Kuopio, Finland. [44] ISER, University of Essex, Colchester CO43SQ, UK. [45] Department of Epidemiology and Public Health, UCL, London, WC1E 6BT, UK. [46] Epidemiology Program, University of Hawaii Cancer Center, Honolulu, Hawaii 96813, USA. [47] MRC Epidemiology Unit, University of Cambridge School of Clinical Medicine, Institute of Metabolic Science, Cambridge CB2 0QQ, UK. [48] Department of Clinical Chemistry, Fimlab Laboratories, Tampere 33520, Finland. [49] Department of Clinical Chemistry, Faculty of Medicine and Life Sciences, University of Tampere, Tampere 33014, Finland.

[50] Department of Twin Research and Genetic Epidemiology, King's College London, London, UK. [51] NIHR Biomedical Research Centre at Guy's and St. Thomas' Foundation Trust, London, UK. [52] Center for Public Health Genomics and Biostatistics Section, Department of Public Health Sciences, University of Virginia, Charlottesville, Virginia 22903, USA. [53] Genetic Epidemiology, QIMR Berghofer Medical Research Institute, Brisbane 4029 , Australia. [54] Institute of Genetic Epidemiology, Helmholtz Zentrum München - German Research Center for Environmental Health, D-85764 Neuherberg, Germany. [55] Department of Medicine I, University Hospital Grosshadern, Ludwig-Maximilians-Universität, D-81377 Munich, Germany. [56] DZHK (German Centre for Cardiovascular Research), partner site Munich Heart Alliance, Munich, Germany. [57] Department of Kinesiology, Faculty of Medicine, Université Laval, Quebec City, Québec, Canada, G1V 0A6. [58] Institute of Nutrition and Functional Foods, Université Laval, Quebec City, Québec, Canada, G1V 0A6. [59] Department of Biotechnology, Institute of Molecular and Cell Biology, University of Tartu, Tartu 51010, Estonia. [60] Department of Social and Health Care, City of Helsinki, Helsinki, Finland. [61] Icelandic Heart Association, Kopavogur, Iceland. [62] Faculty of Medicine, University of Iceland, Reykjavik, Iceland. [63] Department of Medicine, Institute of Clinical Medicine, University of Eastern Finland, 70210 Kuopio, Finland.. [64] Cardiovascular Medicine Unit, Department of Medicine Solna, Karolinska Institutet, Stockholm, Sweden. [65] Center for Molecular Medicine, Karolinska University Hospital Solna, Stockholm, Sweden. [66] Division of Biostatistics, Washington University School of Medicine, St Louis, Missouri, USA. [67] Translational Gerontology Branch, National Institute on Aging, Baltimore, Maryland, USA. [68] Institute for Community Medicine, University Medicine Greifswald, Germany. [69] Department of Cardiology, Leiden University Medical Center, Leiden, The Netherlands. [70] Department of Gerontology and Geriatrics, Leiden University Medical Center, Leiden, The Netherlands. [71] Laboratory of Experimental Cardiology, Department of Cardiology, Division Heart & Lungs, UMC Utrecht, Utrecht, The Netherlands. [72] Department of Epidemiology, University of Groningen, University Medical Center Groningen, The Netherlands. [73] Department of Endocrinology, University of Groningen, University Medical Center Groningen, Groningen, The Netherlands. [74] Divisions of Endocrinology and Genetics and Center for Basic and Translational Obesity Research, Boston Children's Hospital, Boston, Massachusetts 02115, USA. [75] Broad Institute of Harvard and MIT, Cambridge, Massachusetts 02142, USA. [76] Department of Cardiology, University Medical Center Groningen, University of Groningen, Groningen, The Netherlands. [77] Department of Biological Psychology, Vrije Universiteit, Amsterdam, The Netherlands. [78] Behavioural Science Institute, Radboud University, Nijmegen, The Netherlands. [79] Department of Genetics, University of North Carolina, Chapel Hill, North Carolina 27599, USA. [80] Dept Epidemiology and Biostatistics, School of Public Health, Imperical College London, UK. [81] Cardiology, Ealing Hospital NHS Trust, Middlesex, UK. [82] Division of Public Health Sciences, Fred Hutchinson Cancer Research Center, Seattle, Washington, USA. [83] Department of Nutrition, Gillings School of Global Public Health, University of North Carolina at Chapel Hill, Chapel Hill, North Carolina 27599, USA. [84] Department of Medicine, University Medical Center Groningen, University of Groningen, Groningen, The Netherlands. [85] Cardiovascular Health Research Unit, Department of Medicine, University of Washington, Seattle, Washington 98101, USA. [86] Busselton Population Medical Research Institute, Nedlands, Western Australia 6009, Australia. [87] PathWest Laboratory Medicine of WA, Sir Charles Gairdner Hospital, Nedlands, Western Australia 6009, Australia. [88] School of Pathology and Laboraty Medicine, The University of Western Australia, 35 Stirling Hwy, Crawley, Western Australia 6009, Australia. [89] Diabetes and Obesity Research Institute, Cedars-Sinai Medical Center, Los Angeles, California, USA. [90] Clinic for Prosthetic Dentistry, Gerostomatology and Material Science, University Medicine Greifswald, Greifswald, Germany. [91] South Texas Diabetes and Obesity Institute, University of Texas Rio Grande Valley, Brownsville, Texas, USA. [92] Human Genetics Center, The University of Texas Health Science Center, PO Box 20186, Houston, Texas 77225, USA. [93] Medical Genomics and Metabolic Genetics Branch, National Human Genome Research Institute, NIH, Bethesda, Maryland 20892, USA. [94] Department of Pharmacology and Systems Therapeutics, Icahn School of Medicine at Mount Sinai, New York, USA. [95] Department of Pharmacology and Therapeutics, University College Cork, Cork, Ireland. [96] Department of Genetics, Rutgers University, Piscataway, New Jersey 08854, USA. [97] Department of Statistics and Biostatistics, Rutgers University, Piscataway, New Jersey 08854, USA. [98] Usher Institute for Population Health Sciences and Informatics, The University of Edinburgh, Scotland, UK. [99] Imperial College Healthcare NHS Trust, London, UK. [100] Department of Vascular Surgery, Division of Surgical Specialties, UMC Utrecht, Utrecht, The Netherlands. [101] EMGO + Institute Vrije Universiteit & Vrije Universiteit Medical Center, Amsterdam, the Netherlands. [102] Department of Nutrition and Dietetics, School of Health Science and Education, Harokopio University, Athens, Greece. [103] Survey Research Center, Institute for Social Research, University of Michigan, Ann Arbor, Michigan, USA. [104] Robertson Centre for Biostatistics, University of Glasgow, Glasgow, UK. [105] Tropical Metabolism Research Unit, Tropical Medicine Research Institute, University of the West Indies, Mona JMAAW15, Jamaica. [106] Unit of Cardiovascular Epidemiology, Institute of Environmental Medicine, Karolinska Institutet, Stockholm, Sweden. [107] Hypertension and Related Disease Centre, AOU-University of Sassari, Sassari, Italy. [108] Research Unit of Molecular Epidemiology, Helmholtz Zentrum München, German Research Center for Environmental Health, D-85764 Neuherberg, Germany. [109] Institute of Epidemiology II, Helmholtz Zentrum München - German Research Center for Environmental Health, D-85764 Neuherberg, Germany. [110] German Center for Diabetes Research, D-85764 Neuherberg, Germany. [111] Department of Public Health and Clinical Medicine, Section for Nutritional Research, Umeå University, Umeå, Sweden. [112] Laboratory of Epidemiology, Demography, and Biometry, National Institute on Aging, National Institutes of Health, Bethesda, Maryland, USA. [113] Interdisciplinary Center Psychopathology and Emotion Regulation (ICPE), University of Groningen, University Medical Centre Groningen, Groningen, The Netherlands. [114] Department of Psychiatry, Washington University School of Medicine, St. Louis, Missouri, USA. [115] Laboratory of Neurogenetics, National Institute on Aging, Bethesda, Maryland, USA. [116] Division of Genomic Medicine, National Human Genome Research Institute, National Institutes of Health, Bethesda, Maryland 20892, USA. [117] Institute of Medical Sciences, University of Aberdeen, Foresterhill, Aberdeen AB25 2ZD, UK. [118] Generation Scotland, Centre for Genomic and Experimental Medicine, University of Edinburgh, Edinburgh, Scotland. [119] St. Olav Hospital, Trondheim University Hospital, Trondheim, Norway. [120] Interfaculty Institute for Genetics and Functional Genomics, University Medicine Greifswald, Greifswald, Germany. [121] Department of Human Genetics, Wellcome Trust Sanger Institute, Hinxton, Cambridge, UK. [122] School of Medicine and Pharmacology, The University of Western Australia, 25 Stirling Hwy, Crawley, Western Australia 6009, Australia. [123] Department of Cardiovascular Medicine, Sir Charles Gairdner Hospital, Nedlands, Western Australia 6009, Australia. [124] Department of Pediatrics, Tampere University Hospital, Tampere 33521, Finland. [125] Department of Pediatrics, Faculty of Medicine and Life Sciences, University of Tampere, Tampere 33014, Finland. [126] Department of Medical Sciences, Molecular Epidemiology, Uppsala University, Uppsala, 751 85, Sweden. [127] Department of Medicine, Division of Cardiovascular Medicine, Stanford University School of Medicine, Stanford, California 94305, USA. [128] Science for Life Laboratory, Uppsala University, Uppsala 750 85, Sweden. [129] Department of Pulmonary Physiology and Sleep Medicine, Sir Charles Gairdner Hospital, Nedlands, Western Australia 6009, Australia. [130] Department of Physiology, Institute of Neuroscience and Physiology, the Sahlgrenska Academy at the University of Gothenburg, Gothenburg, Sweden. [131] Department of Epidemiology and Biostatistics, MRC–PHE Centre for Environment & Health, School of Public Health, Imperial College London, Norfolk Place, London, UK. [132] Center for Life Course Epidemiology, Faculty of Medicine, University of Oulu, P.O.Box 5000, FI-90014, Oulu, Finland. [133] Biocenter Oulu, University of Oulu, Oulu, Finland. [134] Unit of Primary Care, Oulu University Hospital, Kajaanintie 50, P.O.Box 20, FI-90220, 90029 Oulu, Finland. [135] Department of Medicine, University of Turku, Turku 20520, Finland. [136] Division of Medicine, Turku University Hospital, Turku 20521, Finland. [137] Department of Clinical Physiology, Tampere University Hospital, Tampere 33521, Finland. [138] Department of Clinical Physiology, Faculty of Medicine and Life Sciences, University of Tampere, Tampere 33014, Finland. [139] Clinical and Molecular Osteoporosis Research Unit, Department of Orthopedics and Clinical Sciences, Skåne University Hospital, Lund University, Malmö, Sweden. [140] Department of Medicine and Abdominal Center: Endocrinology, University of Helsinki and Helsinki University Central Hospital, Helsinki FI-00029, Finland. [141] Minerva Foundation Institute for Medical Research, Biomedicum 2U, Helsinki FI-00290, Finland. [142] Department of Public Health, Faculty of Medicine, University of Split, Split, Croatia. [143] Department of Cardiology, Onassis Cardiac Surgery Center, Athens, Greece. [144] Department of Medicine, University of Eastern Finland and Kuopio University Hospital, 70210 Kuopio, Kuopio, Finland. [145] HUNT Research Centre, Department of Public Health and Nursing, Norwegian University of Science and Technology, 7600 Levanger, Norway. [146] Institute

of Biomedicine/Physiology, University of Eastern Finland, Kuopio Campus, Finland. [147] USC-Office of Population Studies Foundation, Inc., University of San Carlos, Cebu City 6000, Philippines. [148] Department of Anthropology, Sociology and History, University of San Carlos, Cebu City 6000, Philippines. [149] Department of Medical Sciences, Cardiovascular Epidemiology, Uppsala University, Uppsala 751 85, Sweden. [150] Li Ka Shing Centre for Health Information and Discovery, The Big Data Institute, University of Oxford, Oxford OX3 7BN, UK. [151] Research Centre for Prevention and Health, the Capital Region of Denmark, Copenhagen, Denmark. [152] Department of Clinical Experimental Research, Rigshospitalet, Glostrup, Denmark. [153] Department of Clinical Medicine, Faculty of Health and Medical Sciences, University of Copenhagen, Copenhagen, Denmark. [154] Translational Laboratory in Genetic Medicine (TLGM), Agency for Science, Technology and Research (A*STAR), 8A Biomedical Grove, Immunos, Level 5, Singapore 138648, Singapore. [155] Department of Public Health and Primary Care, University of Cambridge, Cambridge, UK. [156] Department of Psychology, University of Notre Dame, Notre Dame, USA. [157] Institute of Medical Informatics, Biometry and Epidemiology, Chair of Genetic Epidemiology, Ludwig-Maximilians-Universität, D-81377 Munich, Germany. [158] Department of Medicine, Internal Medicine, Lausanne university hospital (CHUV), Lausanne, Switzerland. [159] Program in Biostatistics and Biomathematics, Fred Hutchinson Cancer Research Center, Seattle, Washington 98109, USA. [160] Molecular Epidemiology, QIMR Berghofer Medical Research Institute, Brisbane, Queensland 4029, Australia. [161] School of Population Health, The University of Western Australia, 35 Stirling Hwy, Crawley, Western Australia 6009, Australia. [162] Department of Respiratory Medicine, Sir Charles Gairdner Hospital, Nedlands, Western Australia 6009, Australia. [163] Institute of Clinical Chemistry and Laboratory Medicine, University Medicine Greifswald, Greifswald, Germany. [164] Institute of Cardiovascular and Medical Sciences, BHF Glasgow Cardiovascular Research Centre, University of Glasgow, Glasgow, Scotland. [165] Research Center for Prevention and Health, Glostrup Hospital, Glostrup, Denmark. [166] Department of Public Health, Faculty of Health Sciences, University of Copenhagen, Copenhagen, Denmark. [167] Centre for Genomic and Experimental Medicine, Institute of Genetics and Molecular Medicine, University of Edinburgh, Edinburgh, Scotland. [168] Department of Clinical Physiology and Nuclear Medicine, Turku University Hospital, Turku 20521, Finland. [169] Research Centre of Applied and Preventive Cardiovascular Medicine, University of Turku, Turku 20520, Finland. [170] Human Genomics Laboratory, Pennington Biomedical Research Center, Baton Rouge, Louisiana, USA. [171] Department of Genetics, Washington University School of Medicine, St. Louis, Missouri, USA. [172] Department of Preventive Medicine, Northwestern University Feinberg School of Medicine, Chicago, Illinois, USA. [173] Division of Cardiology, Brigham and Women's Hospital, Boston, Massachusetts, USA. [174] Istituto di Ricerca Genetica e Biomedica (IRGB), Consiglio Nazionale Delle Ricerche (CNR), Cittadella Universitaria di Monserrato, 09042 Monserrato, Italy. [175] BHF Glasgow Cardiovascular Research Centre, Faculty of Medicine, Glasgow, UK. [176] Laboratory of Genetics, National Institute on Aging, National Institutes of Health, Baltimore, Maryland, USA. [177] Science for Life Laboratory, Karolinska Institutet, Stockholm, Sweden. [178] Division of Angiology, Department of Internal Medicine, Medical University of Graz, Graz, Austria. [179] Department of Molecular Epidemiology, Leiden University Medical Center, Leiden, The Netherlands. [180] Research Unit Hypertension and Cardiovascular Epidemiology, Department of Cardiovascular Science , University of Leuven, Campus Sint Rafael, Kapucijnenvoer 35, Leuven, Belgium. [181] R&D VitaK Group, Maastricht University, Brains Unlimited Building, Oxfordlaan 55, Maastricht, The Netherlands. [182] Institute of Cardiovascular and Medical Sciences, Faculty of Medicine, University of Glasgow, Glasgow, UK. [183] Department of Genetics, University of Groningen, University Medical Center Groningen, Groningen, The Netherlands. [184] Center for Translational Genomics and Population Sciences, Los Angeles Biomedical Research Institute at Harbor/UCLA Medical Center, Torrance, California, USA. [185] Department of Pediatrics, University of California Los Angeles, Los Angeles, California, USA. [186] Department of Public Health Sciences, Stritch School of Medicine, Loyola University of Chicago, Maywood, Illinois 61053, USA. [187] Research Division, Dasman Diabetes Institute, Dasman, Kuwait. [188] Department of Neurosciences and Preventive Medicine, Danube-University Krems, 3500 Krems, Austria. [189] Chronic Disease Prevention Unit, National Institute for Health and Welfare, Helsinki, Finland. [190] Saudi Diabetes Research Group, King Abdulaziz University, Jeddah, Saudi Arabia. [191] Department of Internal Medicine, Erasmus Medical Center, Rotterdam, The Netherlands. [192] School of Nutrition, Université Laval, Laval, Québec, Canada. [193] Department of Public Health and Center for Healthy Aging, University of Copenhagen, 1014 Copenhagen, Denmark. [194] MRC Unit for Lifelong Health and Ageing at UCL, 33 Bedford Place, London WC1B 5JU, UK. [195] Dipartimento di Scienze Farmacologiche e Biomolecolari, Università di Milano, Milan, Italy. [196] Centro Cardiologico Monzino, IRCCS, Milan, Italy. [197] Inserm U-1018, CESP, 94807 Villejuif cedex, France. [198] Geriatric Unit, Azienda USL Toscana centro, Florence, Italy. [199] Lifelines Cohort Study, PO Box 30001, 9700 RB Groningen, The Netherlands. [200] Division of Genetics, Brigham and Women's Hospital, Boston, Massachusetts, USA. [201] Institute for Translational Genomics and Population Sciences, Los Angeles BioMedical Research Institute and Department of Pediatrics, Harbor-UCLA, Torrance, California 90502, USA. [202] Dipartimento di Scienze Biomediche, Universita' degli Studi di Sassari, Sassari, Italy. [203] Sanipedia srl, Bresso (Milano), Italy and Institute of Biomedical Technologies National Centre of Research Segrate, Milano, Italy. [204] Department of Public Health & Clinical Medicine, Umeå University, Umeå, Sweden. [205] Department of Genomics of Common Disease, Imperial College London, London, UK. [206] Carolina Population Center, University of North Carolina at Chapel Hill, Chapel Hill, North Carolina 27516, USA. [207] Department of Psychiatry and Psychotherapy, University Medicine Greifswald, Greifswald, Germany. [208] German Center for Neurodegenerative Diseases (DZNE), Rostock and Greifswald Site, Greifswald, Germany. [209] Department of Preventive Medicine, Norris Comprehensive Cancer Center, Keck School of Medicine, University of Southern California, Los Angeles, California 90089, USA. [210] Durrer Center for Cardiogenetic Research, Amsterdam, The Netherlands. [211] Interuniversity Cardiology Institute of the Netherlands, Utrecht, The Netherlands. [212] Faculty of Med, National Heart & Lung Institute, Cardiovascular Science, Hammersmith Campus, Hammersmith Hospital, Hammersmith Campus, Imperial College London, London, UK. [213] Synlab Academy, Synlab Services GmbH, Mannheim, Germany. [214] Clinical Institute of Medical and Chemical Laboratory Diagnostics, Medical University of Graz, Graz, Austria. [215] Oxford National Institute for Health Research (NIHR) Biomedical Research Centre, Churchill Hospital, Oxford, UK. [216] Department of Biostatistics, University of Liverpool, Liverpool L69 3GL, UK. [217] School of Public Health, University of Adelaide, Adelaide, South Australia 5005, Australia. [218] Laboratory of Clinical Chemistry and Hematology, Division Laboratories & Pharmacy, UMC Utrecht, Utrecht, The Netherlands. [219] Department of Medicine, University of Washington, Seattle, Washington 98195, USA. [220] Department of Epidemiology, University of Washington, Seattle, Washington 98101, USA. [221] Group Health Research Institute, Group Health Cooperative, Seattle, Washington 98101, USA. [222] Department of Epidemiology, School of Public Health and Tropical Medicine, Tulane University, New Orleans, Louisiana, USA. [223] Department of Clinical Physiology and Nuclear Medicine, Kuopio University Hospital, Kuopio, Finland. [224] Division of Population Health Sciences, Ninewells Hospital and Medical School, University of Dundee, Dundee, DD2 4RB, Scotland. [225] Department of Clinical Epidemiology (formerly Institute of Preventive Medicine), Bispebjerg and Frederiksberg Hospital (2000 Frederiksberg), The Capital Region, Copenhagen, Denmark. [226] MRC Integrative Epidemiology Unit, Bristol University, Bristol, UK. [227] Department of Psychiatry Erasmus Medical Center, Rotterdam, The Netherlands. [228] Durrer Center for Cardiogenetic Research, ICIN-Netherlands Heart Institute, Utrecht, The Netherlands. [229] Usher Institute for Population Health Sciences and Informatics, The University of Edinburgh, Scotland, UK. [230] Genetics of Complex Traits, University of Exeter Medical School, RILD Building University of Exeter, Exeter EX2 5DW, UK. [231] European Centre for Environment and Human Health, University of Exeter Medical School, The Knowledge Spa, Truro TR1 3HD, UK. [232] Genetics of Complex Traits, University of Exeter Medical School, University of Exeter, Exeter EX1 2LU, UK. [233] Wellcome Trust Sanger Institute, Hinxton, Cambridge, UK. [234] NIHR Cambridge Biomedical Research Centre, Level 4, Institute of Metabolic Science Box 289 Addenbrooke's Hospital, Cambridge CB2 OQQ, UK. [235] University of Cambridge Metabolic Research Laboratories, Level 4, Institute of Metabolic Science Box 289 Addenbrooke's Hospital, Cambridge CB2 OQQ, UK. [236] Princess Al-Jawhara Al-Brahim Centre of Excellence in Research of Hereditary Disorders (PACER-HD), King Abdulaziz University, Jeddah, Saudi Arabia. [237] Department of Genetics, Harvard Medical School, Boston Massachusetts 02115, USA. [238] Department of Epidemiology, Harvard T.H. Chan School of Public Health, Boston, Massachusetts 02115, USA. [239] Channing Division of Network Medicine, Department of Medicine, Brigham and Women's Hospital and Harvard Medical School, Boston, Massachusetts 02115, USA. [240] Division of Population Health Sciences and Education, St George's, University of London, London SW17 0RE, UK. [241] Netherlands Genomics

Initiative (NGI)-sponsored Netherlands Consortium for Healthy Aging (NCHA). Leiden, The Netherlands. [242] Center for Medical Systems Biology, Leiden, The Netherlands. [243] Institute of Genetic Epidemiology, Helmholtz Zentrum München - German Research Center for Environmental Health, Neuherberg 85764, Germany. [244] Department of Statistics, University of Oxford, Oxford, UK. [245] Mount Sinai School of Medicine, New York 10029, USA. [246] The Mindich Child Health and Development Institute, Icahn School of Medicine at Mount Sinai, New York, New York 10029, USA. [247] Department of Preventive Medicine, The Icahn School of Medicine at Mount Sinai, New York, New York 10029, USA. * These authors contributed equally to this work. ** These authors jointly supervised this work.
‡Deceased.

