## [Peer Review File · Nature Communications]

Reviewers' comments:

Reviewer #1 (Remarks to the Author):

Justice et al. analyzed GWAS data for the detection of novel variants for obesity-related traits. Different to previous approaches smoking behavior, that is known to influence body weight regulation, was taken into account. GWAS data of more than 50,000 current smokers and more than 190,000 non-smokers were used for the analyses. A total of 23 novel loci plus 9 loci with convincing evidence for SNP - smoking interaction were detected. A large number of in silico analyzes (eQTL, expression, epigenetic, PubMed, biological pathway analyzes to name a few) were performed for all newly derived genomic regions. Interesting and partially unexpected new interactions and biological pathways were detected.

Analyses are up to date taking into account a large variety of publically available data bases pertaining to function, biological pathways and epigenetics.

The conclusions are robust, valid and reliable.

Discussion is thoughtful and comprises all relevant aspects.

The approach elegantly shows that taking environmental factors into account might be very useful to detect genetic loci that could not be identified by analysis of the trait under study alone.

Reviewer #2 (Remarks to the Author):

A. Summary of the key results

This study used GWAS data, including approximately 51,000 current smokers and 190,000 nonsmokers, mostly of European descent, to identify novel loci influencing general and central adiposity, as measured by various combinations of weight and height, and waist and hip circumference. In all the authors identified 23 novel genetic loci related to these adiposity indices, and nine loci where the genetic effects appeared to be modified by smoking status. The authors noted that main effects of genetic loci may sometimes be masked by the presence of opposite effects among smokers and non-smokers. Further, some genetic effects may present as main effects, but actually carry stronger influence in smokers than in non-smokers (or vice-versa).

B. Originality and interest: if not novel, please give references

I was very impressed by this comprehensive piece of work which should increase our understanding of the relationship between adiposity and smoking.

C. Data & methodology: validity of approach, quality of data, quality of presentation

The authors used four "approaches" to modelling in their GWAS analysis - (1) only considering main effects, (2) considering main effects in the presence of interaction, (3) considering interaction effects across the whole genome, (4) considering interaction effects only in variants where main effects have already been established

D. Appropriate use of statistics and treatment of uncertainties

I cannot really comment on the biological plausibility of the findings, but I have four main statistical comments which to some extent the authors have already addressed

1. In clinical trials, the presence of subgroup effects, whereby the effect of a treatment may be modified by another factor such as sex or age group, may occur in two different ways, often called qualitative and quantitative interactions. Qualitative interactions occur when the effect of the treatment in one subgroup is in the opposite direction to the other. Pocock et al 2002 (Statistics in Medicine) suggest these are rare and usually implausible. Quantitative interactions occurs when the treatment effect is stronger in one subgroup than another. In searching for such interactions, there are special issues over both Type 1 and Type 2 errors. There is considerably less power to detect interactions than main effects. Also the hunt for interactions may among a plethora of possibilities may lead to false positive findings.

The study of Justice et al seems to find both types of interaction. They frequently refer to reference 9 when Taylor et al, studying a single SNP related to smoking quantity in the CHRNA5-CHRNA3-CHRNA4 locus, showed an unexpected positive effect of the minor allele on BMI among never-smokers, together with the expected inverse effect among smokers. Justice et al appear to reproduce this finding but also show analogous effects among other gene loci. The question is - how plausible are such qualitative interactions, and can we be sure they are not simply chance?

To be fair, Figure 3 helpfully demonstrates considerable power to detect qualitative interactions (if they actually exist), especially for approach 3. Neither approach appears at all powerful to detect quantitative interactions, however. Does this suggests that findings of apparent quantitative interactions are most likely to be chance? The same may be true also of qualitative interactions if their existence is actually very rare.

When carrying out a GWAS on main effects (approaches 1 and 2), the authors use $p < 5 \times 10^{-8}$ and they do the same when searching for interactions in approach 3. Did they consider a double correction when searching for both main effects and interactions?

2. The investigators decided to lump ex-smokers with never smokers into a general "non-smoker" category. While this has the virtue of simplicity, it may miss some nuances of the effects. Taylor et al showed that while opposite effects were seen for current smokers and never-smokers, findings for ex-smokers fell neatly in between (as might be expected, given the in between exposure to smoking that this group would have experienced). I was especially curious as to the effects of R² of the lumping by Justice et al: this would have induced heterogeneity of exposure to smoking, possibly thus reducing the R² for gene vs adiposity trait. The smokers meanwhile are also a heterogeneous group, with a mixture of light and heavy smokers. I think this bears particularly on some results of the enrichment analysis (lines 679-691), where no difference in beta coefficients is discernible between smokers and non-smokers, but differing R²'s are reported.

I also suggest further analysis on novel hits, to see if effects for ex-smokers fall in between those for never-smokers and current smokers, for some of the studies for which the lead authors have rapid access to individual data.

3. Three trait indices are used all of which are complex combinations of anthropometric measures. Even well-known BMI involves a ratio of weight and height². There may well be a residual relationship of this index with height. The authors respond by asserting that no known loci for say height are contiguous with the hits found by the present study. I think this is only an indirect defence (dependent on arbitrary significance levels for GWAS's conducted on height). Especially for WCadjBMI, and especially WHRadjBMI (which is arguably a double adjustment for general adiposity), one wonders if the indices represent artefacts. For WCadjBMI, a sensitivity analysis might involve say log(WC) as dependent variable with log(weight) and height as covariates. I know it takes a long time to run sensitivity analysis when you are dependent on hundreds of co-investigators of the primary studies to

run these. Perhaps the authors have individual data quickly available for a few of the bigger studies to see if novel hits still exhibit directionally similar betas when such a sensitivity analysis is carried out?

4. I was curious at the novel use of four analytic approaches. Approaches 1 and 2 are both looking for main effects, but approach 2 allows for interactions while approach 1 does not. This must mean that if interactions do actually exist, then approach 1 results in undue precision for the main effect. The table at the foot of Figure 1 partly bears this out: fewer loci and fewer novel loci are detected by approach 2 than approach 1. Does approach 1 give some false positive findings for main effects if interactions are present? It would be interesting to see a cross tabulation of effects detected by the two approaches. I imagine that any hits found by approach 2 but not approach 1 is likely to be chance alone.

At the same time, that table at the foot of Figure 1 suggests very few extra hits found in approaches 3 and 4 compared with approaches 1 and 2. This is probably because of the lack of power to detect interactions compared with main effects (see my point 1 above). Also, approach 4 is capable of detecting only quantitative interactions (the sort that are considered more plausible in a clinical trial context), but not qualitative ones.

E. Conclusions: robustness, validity, reliability

Need to check whether some novel hits are really plausible or simply false positives, in view of comments under D above.

F. Suggested improvements: experiments, data for possible revision

See my suggestion of sensitivity analysis for the adiposity indices, especially WCadjBMI (comment D3), and for dividing non-smokers into never and current (comment D2).

G. References: appropriate credit to previous work?

Am not qualified to appraise this fully

H. Clarity and context: lucidity of abstract/summary, appropriateness of abstract, introduction and conclusions

The abstract is nice to read. But the term GXSMK is undefined - I knew what the authors meant but it may not be obvious to all. Also the last sentence, when referring to "genetic susceptibility" - does the directions of effect depend on whether you are discussing the minor or major allele for each individual SNP?

REVIEWERS' COMMENTS:

Reviewer #2 (Remarks to the Author):

I appreciate the thorough further work carried out by the authors to address my previous concerns about Type i error rates.

I have no further comments.

Response to Reviewer Comments: “Genome-Wide Meta-Analysis of 241,258 Adults Accounting for Smoking Behavior Identifies Novel Loci for Obesity Traits”

We thank the two reviewers for their detailed comments, which we feel have led to changes that greatly enhanced our manuscript. We have taken great care to revise our work in line with all of the reviewer’s concerns. Below you will find a detailed response for each point raised and references to any related text corrections or alterations (highlighted in yellow in the attached revised manuscript). To allow for these adjustments to the text and to stay within the manuscript page limits, we have removed some text, tables, and references, as highlighted in the revised draft. Please note that all line numbers referenced below refer to the line numbers contained in the revised draft, as line numbers are not used in the final draft.

REVIEWER #1:

Justice et al. analyzed GWAS data for the detection of novel variants for obesity-related traits. Different to previous approaches smoking behavior, that is known to influence body weight regulation, was taken into account. GWAS data of more than 50,000 current smokers and more than 190,000 non-smokers were used for the analyses. A total of 23 novel loci plus 9 loci with convincing evidence for SNP - smoking interaction were detected. A large number of in silico analyzes (eQTL, expression, epigenetic, PubMed, biological pathway analyzes to name a few) were performed for all newly derived genomic regions. Interesting and partially unexpected new interactions and biological pathways were detected.

Analyses are up to date taking into account a large variety of publically available data bases pertaining to function, biological pathways and epigenetics.

The conclusions are robust, valid and reliable.

Discussion is thoughtful and comprises all relevant aspects.

The approach elegantly shows that taking environmental factors into account might be very useful to detect genetic loci that could not be identified by analysis of the trait under study alone.

RESPONSE:

We thank the reviewer for this appreciation of the novelty of our analytical approaches and results. We hope that our responses to other concerns raised will only strengthen your opinion and appreciation for the proposed manuscript.

REVIEWER #2:

1. Summary of the key results

This study used GWAS data, including approximately 51,000 current smokers and 190,000 nonsmokers, mostly of European descent, to identify novel loci influencing general and central adiposity, as measured by various combinations of weight and height, and waist and hip circumference. In all the authors identified 23 novel genetic loci related to these adiposity indices, and nine loci where the genetic effects appeared to be modified by smoking status.

The authors noted that main effects of genetic loci may sometimes be masked by the presence of opposite effects among smokers and non-smokers. Further, some genetic effects may present as main effects, but actually carry stronger influence in smokers than in non-smokers (or vice-versa).

B. Originality and interest: if not novel, please give references

I was very impressed by this comprehensive piece of work which should increase our understanding of the relationship between adiposity and smoking.

C. Data & methodology: validity of approach, quality of data, quality of presentation

The authors used four "approaches" to modelling in their GWAS analysis - (1) only considering main effects, (2) considering main effects in the presence of interaction, (3) considering interaction effects across the whole genome, (4) considering interaction effects only in variants where main effects have already been established

RESPONSE:

We appreciate that the reviewer has recognized the main strengths and findings of our manuscript. We hope that the responses to the below criticisms and concerns have improved confidence in the results and have only acted to improve your overall evaluation of the manuscript.

D. Appropriate use of statistics and treatment of uncertainties

I cannot really comment on the biological plausibility of the findings, but I have four main statistical comments which to some extent the authors have already addressed

1. In clinical trials, the presence of subgroup effects, whereby the effect of a treatment may be modified by another factor such as sex or age group, may occur in two different ways, often called qualitative and quantitative interactions. Qualitative interactions occur when the effect of the treatment in one subgroup is in the opposite direction to the other. Pocock et al 2002 (Statistics in Medicine) suggest these are rare and usually implausible. Quantitative interactions occurs when the treatment effect is stronger in one subgroup than another. In searching for such interactions, there are special issues over both Type 1 and Type 2 errors. There is considerably less power to detect interactions than main effects. Also the hunt for interactions may among a plethora of possibilities may lead to false positive findings.

The study of Justice et al seems to find both types of interaction. They frequently refer to reference 9 when Taylor et al, studying a single SNP related to smoking quantity in the CHRNA5-CHRNA3-CHRNA4 locus, showed an unexpected positive effect of the minor allele on BMI among never-smokers, together with the expected inverse effect among smokers. Justice et al appear to reproduce this finding but also show analogous effects among other gene loci. The question is - how plausible are such qualitative interactions, and can we be sure they are not simply chance?

To be fair, Figure 3 helpfully demonstrates considerable power to detect qualitative interactions (if they actually exist), especially for approach 3. Neither approach appears at all powerful to detect quantitative interactions, however. Does this suggests that findings of apparent quantitative interactions are most likely to be chance? The same may be true also of qualitative interactions if their existence is actually very rare.

When carrying out a GWAS on main effects (approaches 1 and 2), the authors use $p < 5 \times 10^{-8}$ and they do the same when searching for interactions in approach 3. Did they consider a double correction when searching for both main effects and interactions?

RESPONSE:

We thank the reviewer for these important comments that led us to include simulations to evaluate type 1 error for the various approaches and also to extend our power analyses. We want to note that under the assumption of true biological interaction, the main effect will range between the two stratified effects, which may lead to a loss of power for Approach 1 to find main effects, yet constant type 1 error rates. Also, Aschard et al. 2010 demonstrated increased power under valid type 1 error for Approach 2 given some interaction exists. Therefore,

any hit found by Approach 2 alone should be due to larger power of the joint test that accounts for interaction rather than increased type 1 error rate. In order to illustrate the validity of the approaches with regards to type 1 error, we conducted simulations. We assumed 50,000 smokers and 180,000 nonsmokers and generated 10,000 independent simulated stratum-specific effect sizes under the stratum-specific null hypotheses of “no stratum-specific effects”. We applied the four approaches to the simulated stratum-specific association results and inferred type 1 error of each approach by visually examining QQ plots (new **Supplementary Fig. 7**) and by calculating type 1 error rates (new **Supplementary Table 17**). We observed uninflated QQ plots as well as low type 1 error rates (ranging from 4.53% to 5.24%) at the 5% nominal level for any simulation scenario and across approaches. Since we are employing conservative Bonferroni-corrected significance levels for each approach, this suggests that any positive finding is equally likely to be chance – irrespective of the utilized approach.

As noted by the reviewer, the power of Approaches 3 and 4 to identify interaction effects differs by type of interaction. In order to provide more details, we have extended our analytical power computations to additional scenarios of varying effect sizes (added small and large effects sizes to the already included medium effect size considerations, see newly extended **Figure 3**). Indeed, power to find quantitative interaction is generally lower than finding pure (effect in one subgroup, zero effect in the other) or qualitative (opposite effects) interaction – a circumstance that mirrors our observation of mostly pure and qualitative interactions. Yet, we want to note to the reviewer that approach 4 shows substantial power to find quantitative effects when large genetic effect sizes are involved. For example, the power of approach 4 to identify interaction comprising a large genetic effect in non-smokers ($R^2_{NONSMK}=0.34\%$, comparable to the realistic *FTO* effect on BMI) that is half ($R^2_{SMK}=0.17\%$) or quarter ($R^2_{SMK}=0.09\%$) as large in smokers, was >99% or 53.3%, respectively (new **Figure 3F**).

We sought additional evidence for validation for the top signals identified in the GIANT meta-analysis in an independent study from the UK Biobank (UKBB). We found consistent direction of effect for each of our novel loci and those with evidence of GxSMK interaction. We also meta-analyzed the GIANT and UKBB results and found that the majority of loci remained significant at the conservative Bonferroni-corrected significance levels ($P<5E-8$) for the respective Approach. We have added these validation results to the main text (See below).

We are confident that these additional analyses illustrate that our models are statistically sound, are not the results of inflated Type I error. While our findings are compelling, especially in light of follow-up analyses presented in our paper, only repeated replication and functional assessments of genes underlying these signals can ultimately confirm these association results.

To add further confidence to the future application of our chosen Approaches, we have also added the following text to describe these findings.

We chose to apply two analytical approaches to detect genetic main effects on adiposity traits. To illustrate the validity and utility of applying both approaches, we performed simulations to infer type 1 error (**Online Methods, Supplementary Table 17 and Supplementary Fig 7**) and analytical power computations (**Figure 3**) for Approaches 1 and 2, and a heat map providing the cross-tabulation of P values for Approach 1 and 2 along with Approach 3 examining interaction only (**Supplementary Fig 8**). We demonstrate that the two approaches yield valid type 1 error rates and that Approach 1 can be more powerful to find associations given zero or negligible quantitative interactions, whereas Approach 2 is more efficient in finding associations given some interaction effects.

Validation of Novel Loci

We pursued validation of the associations of our novel and interaction SNPs in an independent study sample of up to 119,644 European adults from the UK Biobank study (**Tables 1-4, Supplementary Table 25, Supplementary Fig 9**). We found consistent directions of effects in smoking strata (for Approach 2 and 3) and in the SNPadjSMK model (Approach 1) for each locus examined (**Supplementary Fig. 13**). For BMI, 3 SNPs were not GWS ($P > 5E-8$) following meta-analysis with our GIANT results, including one of the two variants near *EPAH3* for Approach 1; rs1809420 identified in Approach 4, which lies within a known locus near *ADAMTS7*, remained significant for interaction, but not for SNPadjSMK; and rs336396 near *INPP4B* identified in Approach 3. For WCadjBMI, 3 SNPs were not GWS ($P > 5E-8$) following meta-analysis with our GIANT results, including rs1545348 near *RAI14* for Approach 1; rs4141488 near *GRIN2A* identified in Approach 3; and rs6012558 near *PRNP* identified in Approach 3. For WHRadjBMI, only 1 SNP from Approach 4 was not significant following meta-analysis with our GIANT results, rs12608504 near *JUND*, remained GWS for SNPadjSMK, but was only nominally significant for interaction ($P_{int}=0.013$).

In the Discussion:

There are several challenges in the ability to validate genetic associations that account for environmental exposure. In addition to exposure harmonization and potential bias due to adjustment for smoking exposure, differences in trait distribution, environmental exposure frequency, ancestry-specific LD patterns and allele frequency across studies may lead to difficulties in replication, especially for gene-by-environment studies^{71,72}. Further, the problem of “winner’s curse” (inflated discovery effects estimates) require larger sample sizes for adequate power in replication studies⁷³. Despite these challenges, we were able to detect consistent direction of effect in an independent sample for all novel loci. Some results that did not remain GWS in the GIANT + UKBB meta-analysis still had results that were just under the threshold for significance, suggesting that a larger sample may be needed to confirm these results, and thus the associations near *INPP4B*, *GRIN2A*, *RAI14*, *PRNP*, and *JUND* should be interpreted with caution.

...

Eighteen of these remained significant in our validation with the UK Biobank sample.

Additionally, we edited the abstract as follows:

We identify 23 novel genetic loci, and nine loci with convincing evidence of gene-smoking interaction (GxSMK) on obesity-related traits. We show consistent direction of effect for all identified loci and significance for 18/23 novel and for 5/9 interaction loci in an independent study sample.

2. The investigators decided to lump ex-smokers with never smokers into a general "non-smoker" category. While this has the virtue of simplicity, it may miss some nuances of the effects. Taylor et al showed that while opposite effects were seen for current smokers and never-smokers, findings for ex-smokers fell neatly in between (as might be expected, given the in between exposure to smoking that this group would have experienced). I was especially curious as to the effects of R2 of the lumping by Justice et al: this would have induced heterogeneity of exposure to smoking, possibly thus reducing the R2 for gene vs adiposity trait. The smokers meanwhile are also a heterogeneous group, with a mixture of light and heavy smokers. I think this bears particularly on some results of the enrichment analysis (lines 679-691), where no difference in beta coefficients is discernible between smokers and non-smokers, but differing R2's are reported.

I also suggest further analysis on novel hits, to see if effects for ex-smokers fall in between those for never-smokers and current smokers, for some of the studies for which the lead authors have rapid access to individual data.

RESPONSE:

We agree with the reviewer that it is quite possible that our means of harmonizing the smoking variable introduced heterogeneity into our effect estimates. However, by allowing for this definition of our smoking exposure, we have greatly increased the sample size. Where there are loci with large differences in effects between current and former smokers, there may yet be additional loci to be discovered. In order to determine if our new loci deviated strongly from the expectation of intermediate effects in former smokers, we have examined the association results in some of our larger available studies (ARIC, FRAMHS, FAMHS, KORA, and WHI) with available data on current vs. former smoking status. Additional large datasets would be needed to fully address the question of whether or not smoking intensity plays a role in these associations. Beta estimates from former smokers exhibited intermediate effects between current and never smokers for 20 of the 34 SNPs tested. For the *CHRNA4* locus identified in the Taylor et al. paper, we show similar evidence of direction of effects with opposite direction between current smokers and never smokers and intermediate effect estimates for former smokers. We have included a table below summarizing these meta-analyses results. Given our small stratum-specific sample sizes, and thus lack of power to detect significant associations, these results should be interpreted with caution and are thus not included in our manuscript.

3. Three trait indices are used all of which are complex combinations of anthropometric measures. Even well-known BMI involves a ratio of weight and height². There may well be a residual relationship of this index with height. The authors respond by asserting that no known loci for say height are contiguous with the hits found by the present study. I think this is only an indirect defence (dependent on arbitrary significance levels for GWAS's conducted on height). Especially for WCadjBMI, and especially WHRadjBMI (which is arguably a double adjustment for general adiposity), one wonders if the indices represent artefacts. For WCadjBMI, a sensitivity analysis might involve say $\log(WC)$ as dependent variable with $\log(\text{weight})$ and height as covariates. I know it takes a long time to run sensitivity analysis when you are dependent on hundreds of co-investigators of the primary studies to run these. Perhaps the authors have individual data quickly available for a few of the bigger studies to see if novel hits still exhibit directionally similar betas when such a sensitivity analysis is carried out?

RESPONSE:

We apologize for any confusion caused by the wording of our conditional analyses on known height variants. In fact, we intended to identify cases in which our novel loci from primary meta-analyses were dependent or independent of known height loci. These are highly correlated anthropometric traits which are known to share underlying biological pathways and shared SNP-trait associations (Shungin et al. 2015); therefore we do expect to find that some new loci associated with WC and WHR are known height loci.

However, as the reviewer points out, associations may exist with other loci in our results that have not been reported in the GWAS Catalog. Accordingly, we conducted a lookup of all of our novel variants (lookups for known loci were reported in cross-trait association analysis in Locke et al. and Shungin et al.) in results from a previous GWAS of Height (Wood et al. 2014). Overall, there are few additional variants that may be associated with height, but not previously reported in GWAS examining height (2 for WHRadjBMI and 2 for WCadjBMI with $P < 0.002$ [0.05/24 SNPs]) or in previous cross-trait association efforts.

We have added these lookup results to our **Supplementary Table 18**, along with other previous GWAS lookups, and added the following text to describe these results.

Given the high phenotypic correlation between WC and WHR with height, and established shared genetic associations that overlap across all three of our adiposity traits and height^{1, 2, 68} we would expect additional cross-trait associations between our novel loci and height that may have been missed due to the strict threshold requirements for reporting associations in the GWAS Catalog. Therefore, we conducted a look-up of all of our novel loci in a recent large GWAS meta-analysis of height⁶⁸ to identify additional overlapping association signals (**Supplementary Table 18**). No novel BMI loci were significantly associated with height ($P < 0.002$ [0.05/24 SNPs]). However, there are additional variants that may be associated with height, but not previously reported in GWAS examining height, including 2 for WHRadjBMI near *EYA4* and *TRIB1*, and 2 for WCadjBMI near *KIF1B* and *HDLBP* ($P < 0.002$).

4. I was curious at the novel use of four analytic approaches. Approaches 1 and 2 are both looking for main effects, but approach 2 allows for interactions while approach 1 does not. This must mean that if interactions do actually exist, then approach 1 results in undue precision for the main effect. The table at the foot of Figure 1 partly bears this out: fewer loci and fewer novel loci are detected by approach 2 than approach 1. Does approach 1 give some false positive findings for main effects if interactions are present? It would be interesting to see a cross tabulation of effects detected by the two approaches. I imagine that any hits found by approach 2 but not approach 1 is likely to be chance alone.

At the same time, that table at the foot of Figure 1 suggests very few extra hits found in approaches 3 and 4

compared with approaches 1 and 2. This is probably because of the lack of power to detect interactions compared with main effects (see my point 1 above). Also, approach 4 is capable of detecting only quantitative interactions (the sort that are considered more plausible in a clinical trial context), but not qualitative ones.

RESPONSE:

We thank the reviewer for these valid comments and want to note that under the assumption of true biological interaction, the main effect will range between the two stratified effects. For example, opposite effects between strata may fully cancel out, resulting in a zero main effect. As a consequence and for given interaction, we expect lower power with Approach 1 to find main effects, yet constant type 1 error rates. We do not expect false positive findings for Approach 1 given interaction. Moreover, we want to note that previous work has demonstrated increased power under valid type 1 error for Approach 2 given some interaction (Aschard et al 2010). Therefore, hits found by Approach 2 alone are likely due to larger power of the joint test to account for interaction rather than increased type 1 error rate. To further illustrate this and to improve clarity, we added simulations to infer type 1 error of Approaches 1 and 2 (**Supplementary Table 17** and **Supplementary Fig. 7**), analytical power computations for Approaches 1 and 2 (**Figure 3**), and a heat map providing the cross-tabulation of Approaches 1 and 2 along with Approach 3 (**Supplementary Fig. 8**). We demonstrate that the two approaches yield valid type 1 error rates and that Approach 1 can be more powerful to find associations given zero or very tiny quantitative interactions, whereas Approach 2 is more efficient in finding associations given some underlying interaction effects.

Please see additions to text noted above in the response to concern 1.

E. Conclusions: robustness, validity, reliability

Need to check whether some novel hits are really plausible or simply false positives, in view of comments under D above.

Please see our responses as noted above.

F. Suggested improvements: experiments, data for possible revision

See my suggestion of sensitivity analysis for the adiposity indices, especially WCadjBMI (comment D3), and for dividing non-smokers into never and current (comment D2).

Please see our responses as noted above

G. References: appropriate credit to previous work?

Am not qualified to appraise this fully.

H. Clarity and context: lucidity of abstract/summary, appropriateness of abstract, introduction and conclusions

The abstract is nice to read. But the term GXSMK is undefined - I knew what the authors meant but it may not be obvious to all. Also the last sentence, when referring to "genetic susceptibility" - does the directions of effect depend on whether you are discussing the minor or major allele for each individual SNP?

RESPONSE:

We thank the reviewer for bringing this to our attention. We have edited the abstract to clearly define GXSMK. Additionally, our results are all oriented on the obesity-increasing allele in the smokers. We understand that the interpretation of findings (increasing vs. decreasing) is always relative. In the case, we have chosen this orientation for its epidemiological significance. We have edited the abstract as follows:

“smoking may alter the genetic susceptibility to overall adiposity and body fat distribution.”

Response to Reviewer Comments: “Genome-Wide Meta-Analysis of 241,258 Adults Accounting for Smoking Behavior Identifies Novel Loci for Obesity Traits”

We would like to thank the reviewers for their willingness to review our revised manuscript and we are glad that they have found our revisions have addressed all of their concerns.

REVIEWER #1:

None provided.

No comments.

REVIEWER #2:

I appreciate the thorough further work carried out by the authors to address my previous concerns about Type I error rates.

I have no further comments.

No comments.